# Quantitative transportomics identifies Kif5a as a major regulator of neurodegeneration

Sahil H Shah[1,2,3]*[†], Lucio M Schiapparelli[2†‡], Yuanhui Ma[4], Satoshi Yokota[1], Melissa Atkins[1], Xin Xia[1], Evan G Cameron[1], Thanh Huang[5], Sarah Saturday[2], Catalina B Sun[1], Cara Knasel[1], Seth Blackshaw[5], John R Yates[4], Hollis T Cline[2], Jeffrey L Goldberg[1]

[1]Byers Eye Institute and Spencer Center for Vision Research, Stanford University, Palo Alto, United States; [2]Scripps Research, Neuroscience Department and the Dorris Neuroscience Center, La Jolla, United States; [3]Neuroscience Graduate Program and Medical Scientist Training Program, University of California, San Diego, La Jolla, United States; [4]Scripps Research, Department of Molecular Medicine, La Jolla, United States; [5]Department of Neuroscience, Johns Hopkins University School of Medicine, Baltimore, United States

*For correspondence:
sahilshah90@gmail.com

[†]These authors contributed equally to this work

Present address: [‡]The Department of Cell Biology, Duke University Medical School, Durham, United States

Competing interest: The authors declare that no competing interests exist.

**Abstract** Many neurons in the adult central nervous system, including retinal ganglion cells (RGCs), degenerate and die after injury. Early axon protein and organelle trafficking failure is a key component in many neurodegenerative disorders yet changes to axoplasmic transport in disease models have not been quantified. We analyzed early changes in the protein 'transportome' from RGC somas to their axons after optic nerve injury and identified transport failure of an anterograde motor protein Kif5a early in RGC degeneration. We demonstrated that manipulating Kif5a expression affects anterograde mitochondrial trafficking in RGCs and characterized axon transport in Kif5a knockout mice to identify proteins whose axon localization was Kif5a-dependent. Finally, we found that knockout of Kif5a in RGCs resulted in progressive RGC degeneration in the absence of injury. Together with expression data localizing Kif5a to human RGCs, these data identify Kif5a transport failure as a cause of RGC neurodegeneration and point to a mechanism for future therapeutics.

## Editor's evaluation

This study uses unbiased proteomics to determine which proteins are transported along the axon transportome in the context of optic nerve injury. Kif5a is identified as the most significantly down-regulated protein in the transportome after injury. Knocking out Kif5a resulted in degeneration of the axon, suggesting that Kif5a is crucial for maintaining a healthy optic nerve. Further, optic nerve analyses of the Kif5a KO vs the control upon injury identified defective mitochondrial transport and defective hRNP transport that has been identified as a cause of neurodegeneration in ALS upper motor neurons.

## Introduction

Adult mammalian central nervous system neurons often undergo cell death and axon degeneration after injury and in disease. Neurodegenerative diseases such as glaucoma and acute injuries such as after optic nerve trauma may result in irreversible retinal ganglion cell (RGC) loss. Mechanisms underlying this degeneration include intracellular events such as acute calcium influx into the axon

(*Knöferle et al., 2010*), loss of constitutive axon survival factors like Nmnat2 (*Gilley et al., 2015*), inhibition of trophic survival and growth pathways, as with suppressors of cytokine signaling (e.g. Socs3) and ciliary neurotrophic factor (CTNF) signaling or Dusp14 and mitogen-activated protein kinases (*Cai et al., 1999*; *Zhou et al., 2005*; *Leaver et al., 2006*; *Park et al., 2010*; *Galvao et al., 2018*), and neuron-extrinsic degenerative signaling, such as from reactive astrocytes (*Liddelow et al., 2017*) and microglia. Major strides in understanding neurodegeneration in neurons including RGCs have been made in recent years, including transcriptomic (*Yasuda et al., 2016*), proteomic (*Belin et al., 2015*), and metabolomic (*Sato et al., 2018*) cellular changes.

Axon transport failure has been pathophysiologically linked to RGC death in glaucoma, as well as to other neurodegenerative diseases such as amyotrophic lateral sclerosis (ALS) and Alzheimer's disease (*Zhang et al., 2004*; *Nicolas et al., 2018*). While changes described in transcriptional and gene-regulatory networks in neurons can be a proxy for changes in a neuron's or an axon's molecular components, it does not capture dynamics in expression or localization of proteins. Proteomics experiments in RGC somas have identified axon injury-induced pathways (*Belin et al., 2015*), but it is not known how molecular trafficking to the damaged optic nerve changes after injury. Unbiased temporal and spatial quantification of protein transport in vivo has presented technical challenges, including the overall low proportion of transported proteins, and the need to purify target axon proteins from surrounding mixed cellular populations.

We previously developed a mass spectrometry-compatible technique for characterizing the axon transportome in the visual system (*Schiapparelli et al., 2019*). Here, we adapt this technique to quantify changes in the axon transportome after optic nerve injury. We identify transport failure of a kinesin-1 motor protein, Kif5a, which we find expressed in human RGCs and also implicated in genetics of neurodegeneration outside the visual system (*Tessa et al., 2008*; *Nicolas et al., 2018*). We use transportome proteomics to identify proteins whose transport is dependent on Kif5a, and finally we validate the importance of Kif5a for adult RGC survival in vivo. Together this study adapts novel methods to identify Kif5a transport deficits as a contributor to RGC death after injury and offers new insights into kinesin motor cargo specificity.

## Results

### Pulsed NHS-biotin labels proteins for differential immunofluorescence and proteomics

After optic nerve crush in rats, 65% of RGCs die within 1 week, while less than 1% die after 1 day (*Sánchez-Migallón et al., 2016*). We were interested in identifying potential differences in axonally transported proteins after axon damage, but before cell death. In previous studies we used repeated daily intravitreal injections of *N*-hydroxysuccinimidobiotin (NHS-biotin) over a week in adult rats for steady-state mass spectrometry-based quantification of anterogradely transported proteins in the optic nerve (*Schiapparelli et al., 2014*; *Schiapparelli et al., 2019*). NHS-biotin covalently binds primary amines on proteins, principally lysines and N-terminal amino acids, adding a biotin group (*Figure 1a*), which is identifiable by the specific shift in peptide molecular weight in mass spectrometry or by immunodetection. Here, we adapted our NHS-biotin protocol to a 24 hr window to provide improved temporal resolution, favoring it over two alternative methods tested (*Figure 1—figure supplement 1*), consistent with our goal to study early changes after optic nerve crush. Animals received two intravitreal injections, the first at the same time as optic nerve crush or sham surgery and a second 21 hr later, 3 hr before tissue collection, to capture both fast and slow axon transport. Although a variety of retinal proteins are labeled by intravitreal injection, dissection of the optic nerve isolates only proteins which have been transported from RGC somas into RGC axons (*Figure 1b*). Biotinylated, transported proteins were visualized with tyramide signal amplification in the control optic nerve and region of the injured optic nerve proximal to the crush site (*Figure 1c*). Notably, there was no biotin signal distal to the crush, confirming the disruption of protein transport and lack of biotin transfer out of axons after injury. Together these data indicate that this protocol results in protein labeling within RGC axons in the optic nerves in both the control and injury conditions, allowing targeted isolation and investigation of changes in RGC proteins transported into the optic nerve after injury.

We used tandem mass spectrometry (MS/MS) combined with direct detection of biotin-containing tags enrichment to selectively isolate and identify biotinylated peptides. Directly measuring biotin

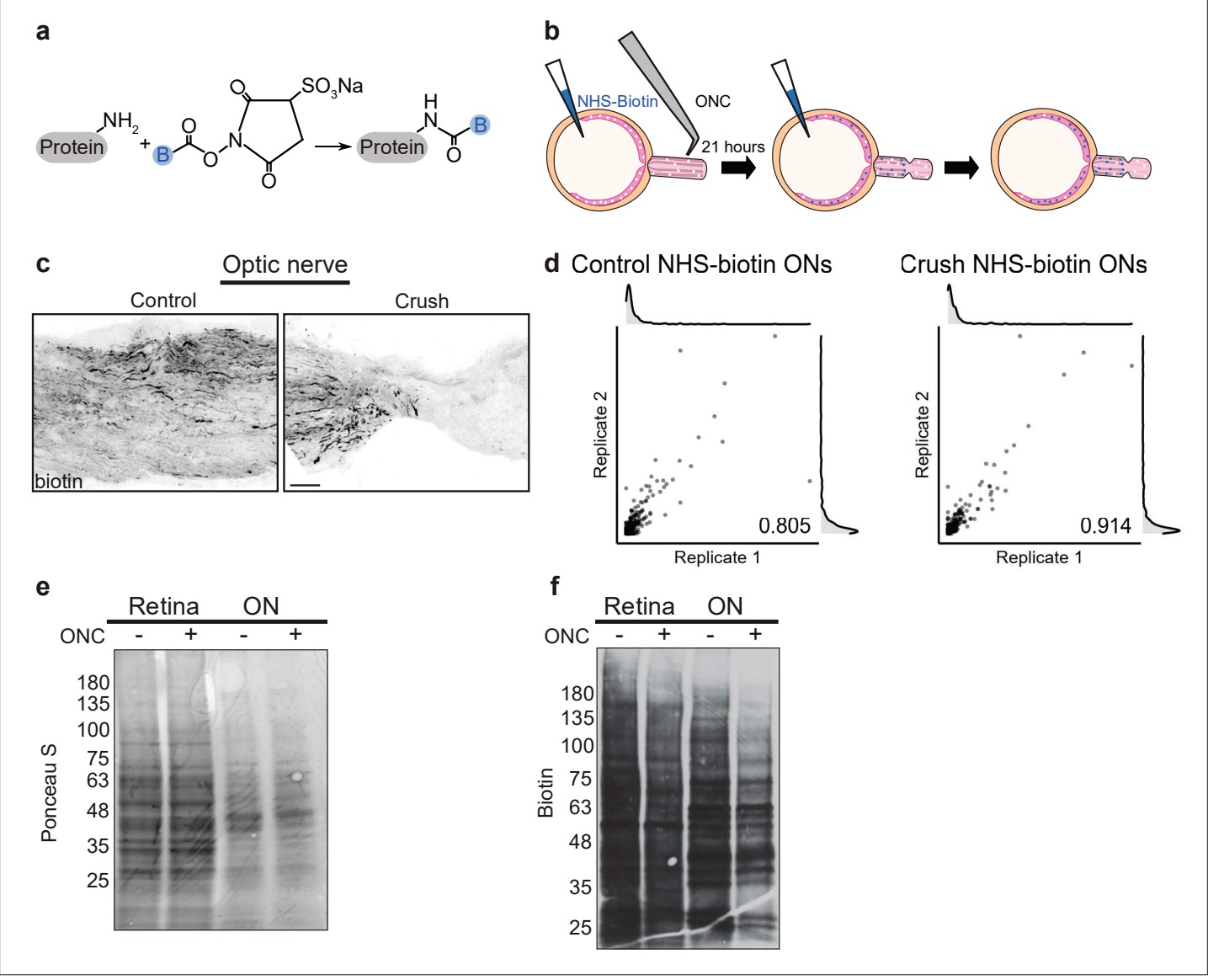

**Figure 1.** Pulsed *N*-hydroxysuccinimidobiotin (NHS-biotin) labels and quantifies protein transport after injury. (**a**) NHS-biotin (biotin group in blue) reacts with primary amines adding a defined mass shift to tagged proteins. (**b**) NHS-biotin (blue) was injected before injury with a repeat injection 21 hr later. Biotinylated proteins in retinal ganglion cells (RGCs) transport down intact RGC axons in the optic nerve. The retinas and optic nerves were collected for tissue processing after 3 hr. (**c**) No biotinylated proteins were detected by immunofluorescence distal to the crush site. Scale bar, 100 μm. (**d**) Pearson's R correlation of proteomic replicates of control or optic nerve crush (ONC) samples (R = 0.805 and 0.914, respectively). (**e**) Ponceau S stain of protein from biotinylated retinas and optic nerves, with and without ONC, showing equivalent total protein. Molecular ladder labeled on left of blot. (**f**) Western blot of the same samples run in parallel, probing for biotin. ONC decreased biotinylated proteins to a greater extent in ON compared to retina. Molecular ladder labeled on left of blot.

The online version of this article includes the following source data and figure supplement(s) for figure 1:

**Source data 1.** Proteomic quantification of transportome changes after ONC from *Figure 1b*.

**Source data 2.** Raw western blots from *Figure 1*.

**Figure supplement 1.** Alternative protein labeling strategies.

**Figure supplement 1—source data 1.** Raw western blots from *Figure 1—figure supplement 1*.

mass shifts on lysine residues in peptides with MS/MS facilitates direct quantification of biotinylated proteins with improved specificity (*Schiapparelli et al., 2014*). To minimize variability and increase protein yield, we combined eight optic nerves of either control or injury conditions for each MS sample. During dissection, we included optic nerve fragments from the end of the orbit to the chiasm from both injured and sham operated samples to ensure comparable tissue processing. A total of six samples, three each of control or optic nerve crush, were run independently through MS/MS. We detected 206–300 transported proteins with a range of detection of expression up to 300-fold normalized spectral abundance factor (NSAF) values. To assess reproducibility of the quantitative proteomics in each of these conditions, we used pairwise correlation of NSAF values which showed high correlation between replicates of both control and crush samples (*Figure 1d*, R = 0.805 and R = 0.914, respectively). These experiments show that reducing the labeling window to 24 hr and comparing changes in NSAF values allow relative quantification between two conditions with high correlation between replicates, even in this tight temporal window before widespread apoptosis after optic nerve injury.

## Quantitative transportomics identifies specific proteins whose transport differs after injury

To ask whether protein transport into the optic nerve changed after injury, we measured the total protein abundance in control and injured retinas and optic nerve by electrophoresis, western blotting, and Ponceau S membrane staining. Total protein content was equivalent in optic nerve crush and control retinas and optic nerves, as expected given the short time between injury and tissue collection (*Figure 1e*). However, when we compared the biotinylated portion of total protein in crush or control conditions, we found a decrease in axon-transported protein after injury (*Figure 1f*), consistent with previous studies of total transport after injury (*Quigley et al., 1979*; *McKerracher et al., 1990*).

To test transport changes of individual proteins, we probed for Sncb, Gap43, and Arf3, neuronal proteins known to be transported down the optic nerve (*Leon et al., 2000*; *Schiapparelli et al., 2019*; *Figure 2a*). Four rat optic nerves were labeled and homogenized using the same protocol as above for proteomics, and then biotin-tagged proteins were immunoprecipitated with neutravidin-bound agarose beads. We compared retinal and optic nerve total proteins (input), as well as the biotinylated fraction pulled down by immunoprecipitation (IP), with and without injury. In the retina, there was little to no change in total protein levels (input, *Figure 2a*) or in the biotinylated fraction (IP, *Figure 2a*) in any of the three proteins, suggesting the abundance of these proteins is equivalent in the retinal cell bodies at this early time point after optic nerve injury. Sncb and Arf3 but not Gap43 showed significant decreases in transport into the optic nerve after injury (IP + vs. – crush, *Figure 2a*), and this decrease was significant even when normalizing to the decrease in total transported protein as captured in *Figure 1f* (normalized $\log_2$FC = −2.2 for Sncb, −1.3 for Arf3, n = 3). This suggests that deficits in transport of specific proteins occur within the first 24 hr after optic nerve injury before changes in total protein can be detected. These data exemplify the need for axon-specific, temporally restricted protein labeling to accurately assess early axon transportome changes in the degenerative process.

We then examined the transportome changes after optic nerve crush using mass spectrometry. We normalized relative abundance of detected proteins within each condition to determine whether specific proteins are transported down the axon in greater or lesser amount relative to the global transport rate, which was decreased. Individual protein transport was quantified using NSAF in injured or control conditions across three replicates containing eight optic nerves each (complete data in *Figure 1—source data 1*). The relative log-normalized fold change of anterogradely transported proteins from RGC bodies to their axons in the optic nerve was visualized on a volcano plot (*Figure 2b*). Because mass spectrometry undersamples the total proteome and thus we did not want to filter too stringently, we analyzed proteins that had a $\log_2$ fold change of greater than 1.5 of relative abundance in the transportome, or a p-value < 0.05, or both. This identified several proteins that fell into three ontologies of identified function: cytoskeletal (Tmsbx, Tcp1, Tuba1a, Tuba1b, Tuba3a, Nefl, and Ina), protein synthesis (Impact), and protein transport (Kif5a). In the protein transport ontology, transport of Kif5a, a neuronal kinesin heavy chain motor protein, was significantly decreased. These data indicate that analyzing the dynamic axon transportome after optic nerve injury identifies acute changes in several protein synthesis and transport homeostatic mechanisms after axon injury.

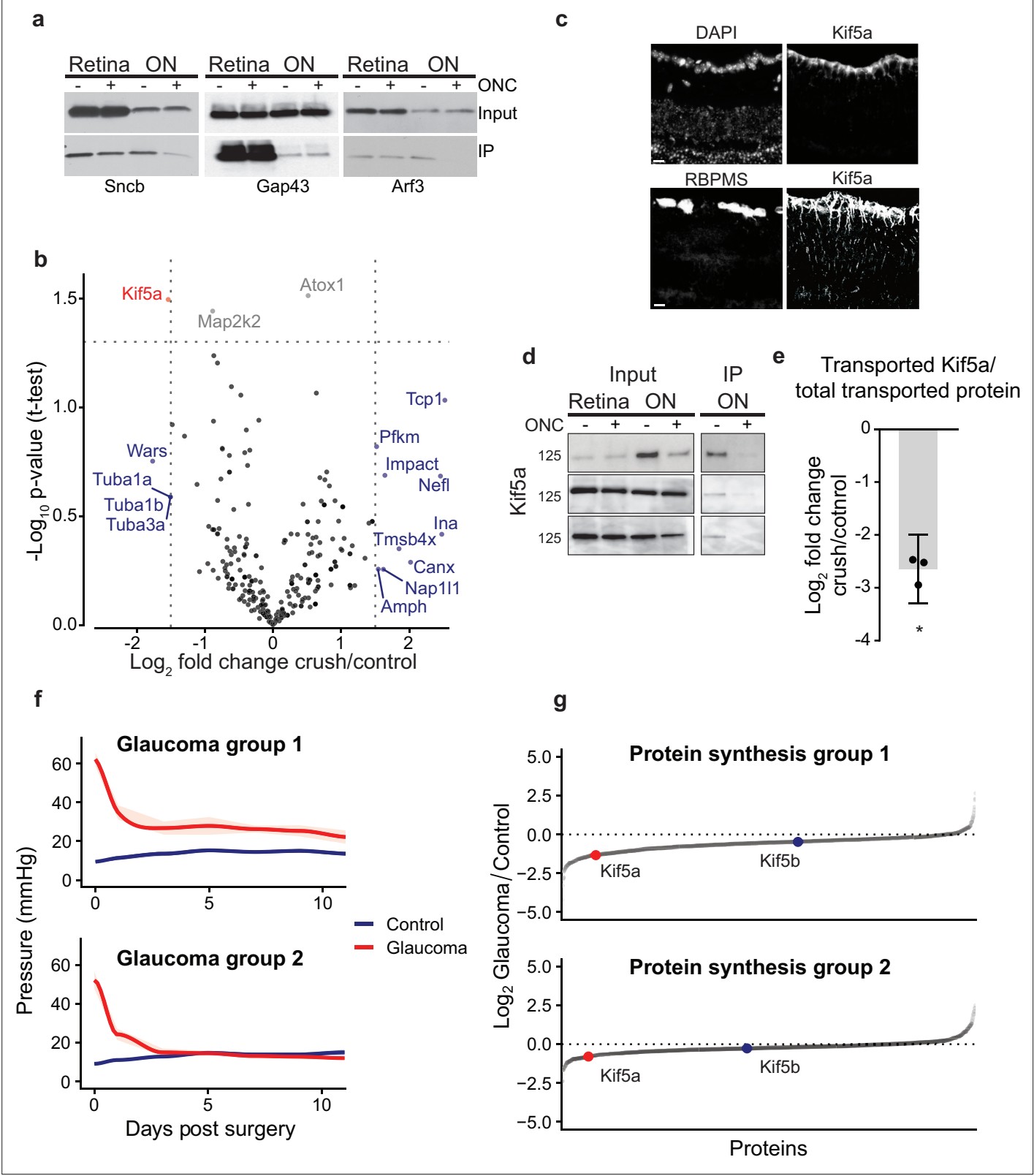

**Figure 2.** Quantitative analysis of proteomics after injury identifies downregulation of Kif5a synthesis and transport after injury. (**a**) Immunoprecipitation (IP) of biotinylated protein from retinal and optic nerve samples, probed for Sncb, Gap43, or Arf3. Input of each sample on top row, subsequent IP of each sample on bottom row, for either optic nerve crush (ONC) samples or sham surgeries. (**b**) Volcano plot comparing biotinylated proteins from control versus ONC nerve samples. Normalized spectral abundance factor (NSAF) values for each sample type were averaged across three replicates.

*Figure 2 continued on next page*

Figure 2 continued

Proteins with an absolute fold change greater than 1.5, p-value less than 0.05, or both, are colored blue, gray, and red respectively. (**c**) Retinal cross-sections immunostained with Kif5a and DAPI showing localization of Kif5a in the retinal ganglion cell (RGC) layer (top) and RGC-specific marker RBPMS and Kif5a showing co-localization of Kif5a with RGCs (bottom). Scale bar, 100 µm. (**d**) Three replicates of biotin IPs probed with an antibody against Kif5a. Inputs are on the left. The IP was stripped and re-probed for biotin for a measurement of total biotinylated protein pulled down (not shown). (**e**) Quantification of the change in transported Kif5a compared to the change in total transported protein after ONC. One-sample, two-tailed t-test, p = 0.003, n = 3. The bar height and error bar represent the mean and 95% confidence interval respectively. ONC = optic nerve crush. (**f**) Average intraocular pressure after induction of glaucoma over time in two groups, one with sustained pressure elevation and one with only initial pressure elevation. (**g**) Quantitative changes in retinal protein synthesis 3 weeks after induction of glaucoma. 1143 and 1457 proteins were quantified in replicates 1 and 2, respectively. Log$_2$ transformed normalized spectral ratios of glaucoma/control samples were plotted from lowest to highest. Newly synthesized Kif5a, detected by BONCAT and quantitative mass spectrometry, was decreased in both replicates, while Kif5b synthesis was unaffected.

The online version of this article includes the following source data and figure supplement(s) for figure 2:

**Source data 1.** Raw western blots for *Figure 2*.

**Figure supplement 1.** Motor protein transport changes after injury and synthesis changes after glaucoma.

**Figure supplement 1—source data 1.** Proteomic quantification of changes in newly synthesized proteins 3 weeks after glaucoma from *Figure 2— figure supplement 1b*.

We confirmed Kif5a expression is localized to RGCs in the mouse retina using immunohistochemistry (*Figure 2c*). We validated the mass spectrometry identification of Kif5a's reduced axon transport by biotin IP and quantitative western blot (*Figure 2d*). We normalized biotin-tagged Kif5a to each sample's total biotin-tagged protein (similar to that shown in *Figure 1f*), and compared injury versus control, which resulted in a significant decrease (about 2.5-fold) in Kif5a transport relative to the decrease in total axon transport after injury (*Figure 2e*), similar to the relative decrease detected by mass spectrometry. Importantly, we saw an equal amount of Kif5a in the retina with and without injury, suggesting that the cause of decreased axon transport is not due to changes in Kif5a abundance in the retina, but in fact reflects a specific anterograde transport deficit. Thus, both western blotting and biotinylated peptide-based mass spectrometry of the axon transportome corroborate Kif5a's reduced axon transport after optic nerve injury.

Kif5a, the only protein that met both significance and magnitude of change criteria, is a motor protein in the 45-member kinesin superfamily (*Hirokawa et al., 2009*). Kinesin-1 motors are composed of two heavy chain and two light chain proteins, where the heavy chains are any of Kif5a, Kif5b, and Kif5c. Of these, Kif5a and Kif5c are neuron-specific, and previously described to be expressed in RGCs (*Rahman et al., 1999*; *Butowt and von Bartheld, 2007*). When all motor proteins in our transportome data were examined (*Figure 2—figure supplement 1a*), two findings stood out. First, another kinesin heavy chain, Kif5b, did not change in relative transport after injury, suggesting specificity in kinesin isoform transport failure. Second, transport of Klc1, a light chain partner of the heavy chain Kif5a, also was not reduced to the same extent as Kif5a. As kinesin and dynein work in tandem to regulate cargo transport (*Hendricks et al., 2010*), we examined whether there may be similar changes to dynein subunits. We did not discover any significant change in transport of both dynein subunits identified in our transportome, although more work may be needed to explore this question.

## Synthesis of Kif5a, but not Kif5b, is decreased in glaucomatous injury

We also explored changes in Kif5a synthesis after axon injury expanding our investigation to another optic neuropathy, the circumlimbal suture model of glaucoma (*Liu et al., 2015*). Glaucoma is the leading cause of irreversible blindness worldwide and is similarly due to the dysfunction and death of RGCs and their axons. We used the noncanonical amino acid azidohomoalanine (AHA), which is incorporated into newly synthesized proteins in place of endogenous methionine during protein translation (*Schiapparelli et al., 2014*; *McClatchy et al., 2015*), to selectively quantify newly synthesized proteins over the same 24 hr time period as used in the previous experiments. Three weeks after induction of glaucoma (*Figure 2f*), AHA was injected twice in a 24 hr period as in the experiments described above. After tissue collection, AHA was tagged with biotin using click chemistry, and changes in newly synthesized proteins were quantified with MS/MS. In the retina, 43 proteins showed significant increased synthesis rates and 30 proteins showed significant decreased synthesis rates at this time point in glaucomatous optic neuropathy, normalized to controls. Among these, Kif5a showed decreased synthesis when normalized to the total newly synthesized proteome after induction of

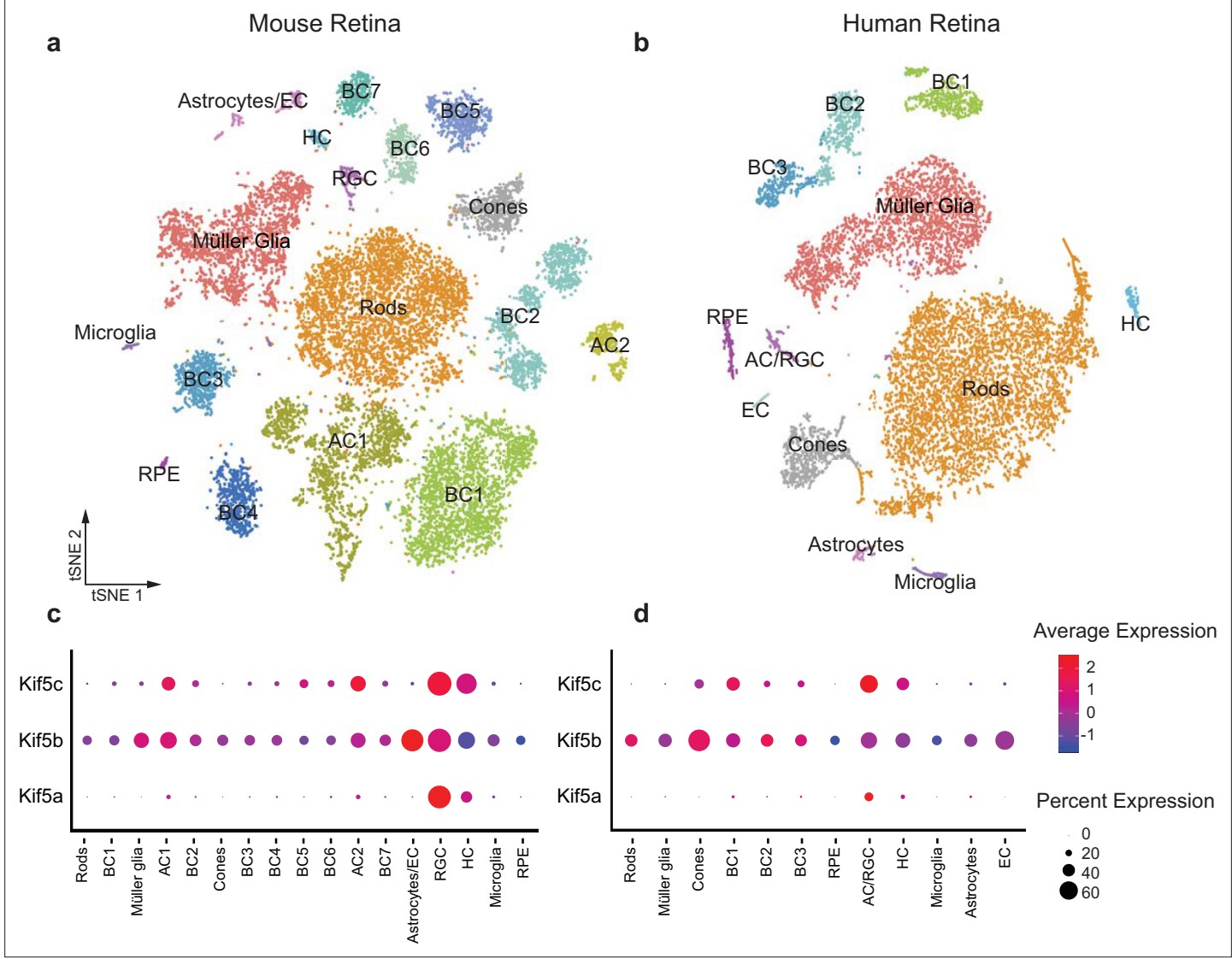

**Figure 3.** *Kif5* expression pattern in the retina is conserved from rodents to humans. (**a,b**) TSNE plot of cell clustering of single-cell RNA sequencing (scRNA-seq) from either an adult mouse retina or an adult human retina. Clusters defined by canonical cell-specific markers. (**c,d**) Expression of *Kif5* genes across retinal cell types in either mouse or human retina. *Kif5a* shows highest levels of enrichment in the putative retinal ganglion cell (RGC) clusters in both species, followed by *Kif5c* enrichment. *Kif5b*, conversely, is more ubiquitous across the retina in both species, displaying both conservation of expression patterning across species and divergence of patterning among *Kif5* family members. EC = endothelial cells, BC = bipolar cells, AC = amacrine cells, HC = horizonal cells, RPE = retinal pigment epithelium. Human scRNA-seq data available at GEO: GSE138002 and mouse scRNA-seq data available at GEO: GSE135406.

glaucoma, while Kif5b was relatively unchanged (***Figure 2g***, ***Figure 2—figure supplement 1b***, complete data in ). Together with the acute optic nerve trauma model, these data demonstrate the specificity of changes in synthesis and transport of Kif5a, as compared to other Kif5 isoforms, in both acute and chronic models of optic nerve disease.

## Kif5 expression in human RGCs

Given the specificity of Kif5a transport failure into RGC axons after optic nerve injury, we wished to explore the expression of *Kif5* mRNA isoforms in RGCs and other retinal cell types. We used single-cell RNA sequencing (scRNA-seq) data from both adult mouse and adult human retinas to determine retinal distribution of kinesin motors and conservation of patterning across species (***Getter et al., 2019***). We were able to cluster retinal cells into constituent cell types, including a putative RGC cluster (***Figure 3a–b***). We found that *Kif5a* was enriched in the amacrine cell/RGC cluster of cells compared

to other retinal cell types in both mouse and human, whereas *Kif5b* was expressed more ubiquitously across retinal cell types (*Figure 3c–d*). The conservation of this RGC-enriched expression pattern for both mouse and human retinas further supports a model of kinesin specificity and function.

## Transportomics of Kif5a knockout in RGCs

Given our finding of an association of early Kif5a transport decrease with optic nerve injury, we sought to understand the importance of Kif5a, and its associated cargoes, in RGC function. Discovering specific kinesin-dependent cargoes, for example, by mass spectrometry has not been undertaken, so we hypothesized that characterizing the optic nerve transportome with and without Kif5a should yield Kif5a-dependent cargo localization. For this approach, we used homozygous conditional floxed *Kif5a* adult mice (*Figure 4—figure supplement 1*) as Kif5a knockout mice have seizures and die soon after birth (*Xia et al., 2003*; *Nakajima et al., 2012*). We delivered either cre-recombinase- or GFP-expressing adeno-associated virus (AAV) intravitreally to specifically knock out Kif5a in adult RGCs. One month after injection, there was almost a complete loss of Kif5a in the retina, and a complete loss in the optic nerve, with neither loss nor compensatory upregulation of Kif5c (*Figure 4a*). We confirmed this loss by immunofluorescence for Kif5a at 1 and 4 months after cre-mediated knockout (*Figure 4b*).

We next explored the Kif5a-dependent transportome. As cre-recombinase requires additional time for expression, recombinase activity, and degradation of existing Kif5a protein, we used a longer time course (3 weeks) to identify the steady-state changes in transport due to Kif5a loss (*Ahmed et al., 2004*). In uninjured mice 3 weeks after intravitreal viral cre injection, we injected NHS-biotin intravitreally three times, each 24 hr apart, to assess changes in the axon transportome using MS/MS as above. This approach identified several proteins with decreased optic nerve axon transport after loss of Kif5a (*Figure 4c*), a number of which are implicated in neuronal injury response and neurodegeneration (complete data in *Figure 4—source data 1*). Bin1, for example, was identified as an Alzheimer's disease locus through GWAS studies (*Seshadri et al., 2010*), and localizes to the axon initial segment and nodes of Ranvier acting downstream of c-myc, an injury-response transcription factor in RGCs (*Belin et al., 2015*). Sh3gl2, also known as endophilin A1, is enriched in synaptic terminals and interacts with synaptic proteins such as synaptojanin and regulates clathrin-mediated synaptic endocytosis (*Milosevic et al., 2011*). We asked whether the proteins whose transport depends on Kif5a expression interact with each other using STRING and were able to group many of these together into three protein-protein interaction networks (*Figure 4d*). Kif5a has already been directly tied to a few of these proteins (*Ishida et al., 2015*), and the presence of potential complexes of non-transported proteins many represent as yet undiscovered functions of Kif5a. Thus these data identify a network of proteins whose transport depends on Kif5a in RGC axons as well as more broadly establish an approach to identify the transportome of motor proteins in vivo.

## Kif5a transport disruption affects anterograde mitochondrial transport in RGCs

We next compared the changes in protein transport seen in optic nerve crush versus Kif5a knockout. We do not expect these entire datasets to correlate given the difference in injury, time course, and model organism, and indeed they do not. However, when we specifically asked how deficits in protein transport after optic nerve crush compared to deficits in protein transport after Kif5a knockout, we find a group of proteins that fulfill both parameters. We focused on protein transport that decreased in both conditions as we hypothesize transport failure contributes to neurodegeneration in both conditions (complete data in *Figure 5—source data 1*). Using Gene Ontology to cluster all proteins that decrease in transport in both optic nerve crush and Kif5a knockout, we discovered a significantly overrepresented number of mitochondrial-associated genes in common (*Figure 5a–b*). With this finding, we hypothesized that in both Kif5a transport failure after optic nerve crush and in Kif5a knockout that mitochondrial transport may be specifically downregulated. We tested this hypothesis in primary cultures of RGCs and measured the effect on mitochondrial transport through live cell imaging. AAV2-mediated Kif5a knockdown significantly reduced the average velocity of non-stationary mitochondria compared to an AAV2 scrambled shRNA virus, as shown in kymographs of mitochondrial transport (*Figure 5c–d*). By quantifying the number of mitochondria transported in each direction, we found a significant reduction in anterograde transport (*Figure 5e*). Together, these findings support a model

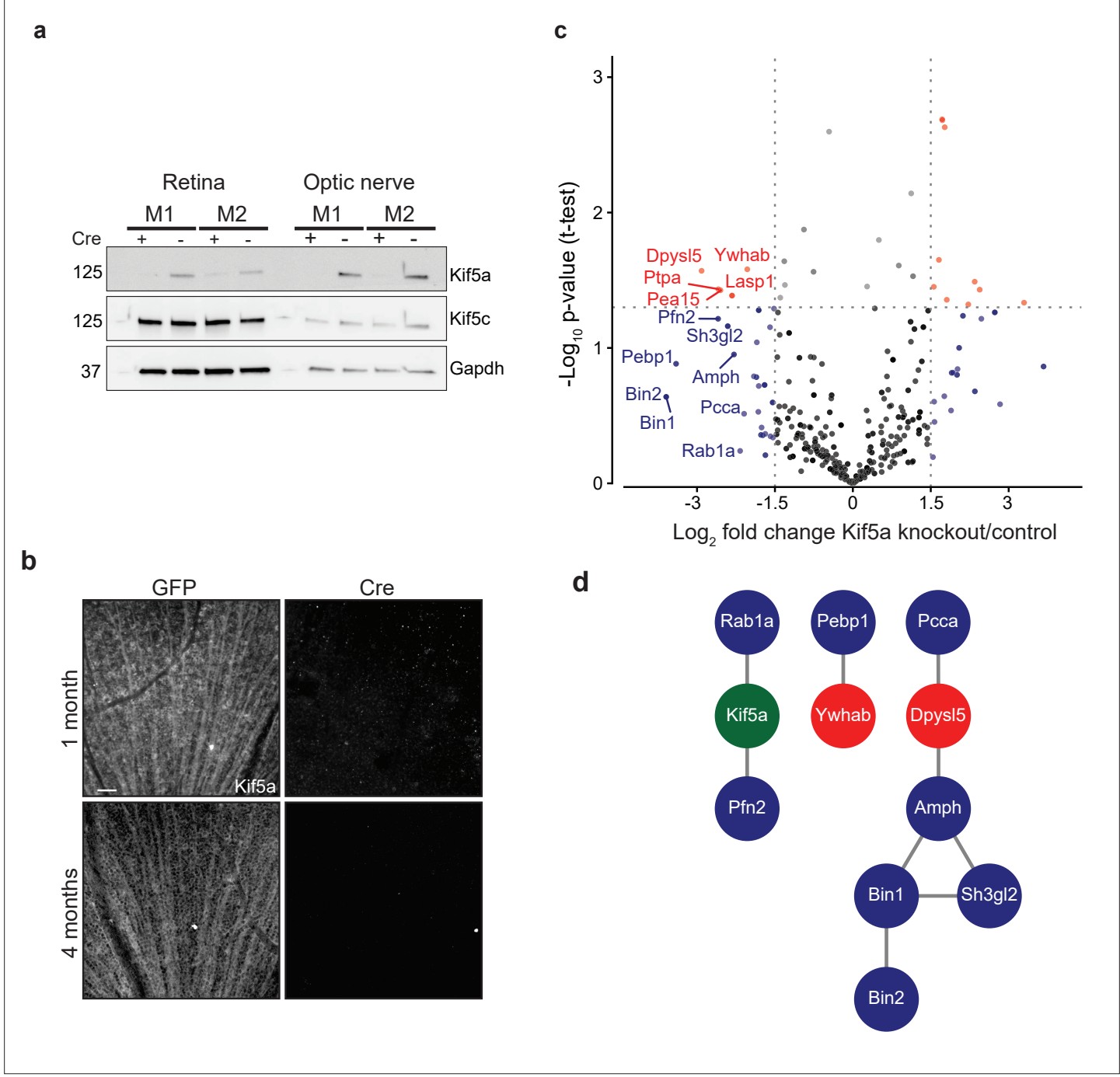

**Figure 4.** Transportomics of Kif5a knockout (KO). (**a**) Western blot validation of Kif5a KO 1 month after intravitreal cre-recombinase injection was apparent in whole retina and optic nerve lysates. Kif5c, a related kinesin, was unaffected by the loss of Kif5a. Cre-positive lanes are designated with a '+', cre-negative lanes are designated with a '-'. Molecular weight on left of blots. (**b**) Wholemount retinas immunostained with an antibody against Kif5a show immunopositive retinal ganglion cell (RGC) axons projecting toward the optic nerve head. Loss of kif5a immunopositive fibers is detected 1 month and is even more pronounced at 4 months after adeno-associated virus (AAV)-cre injection. Scale bar, 50 µm. (**c**) Volcano plot comparing biotinylated proteins in the optic nerve 3 weeks after intravitreal AAV-GFP compared to AAV-Cre-GFP. Proteins with a $\log_2$ fold change decrease of greater than 2 after KO are shown in blue, and those which additionally had a p-value less than 0.5 are shown in red. (**d**) STRINGdb interaction diagram for all labeled proteins plus Kif5a create three networks incorporating 11/14 proteins (blue = decrease transport only, red = significantly decreased transport).

The online version of this article includes the following source data and figure supplement(s) for figure 4:

**Source data 1.** Proteomic quantification of transportome changes in Kif5a knockout from *Figure 4c*.

**Source data 2.** Raw western blots for *Figure 4*.

*Figure 4 continued on next page*

*Figure 4 continued*

**Figure supplement 1.** Genotype confirmation of conditional Kif5a knockout (KO) mice.

**Figure supplement 1—source data 1.** Raw genotyping blots for *Figure 4—figure supplement 1*.

of anterograde mitochondrial trafficking failure in both Kif5a knockout and following optic nerve crush, suggesting a shared biology between injury and loss of Kif5a.

We also tested whether exogenous expression of Kif5a could increase anterograde mitochondrial trafficking. Primary RGCs treated with an AAV2 Kif5a expression virus demonstrated significantly increased average velocity (*Figure 5f–g*). We also found a significant increase in the fraction of anterogradely moving mitochondria compared to control (*Figure 5h*). Thus Kif5a expression is necessary for and can promote anterograde mitochondrial transport in mammalian RGCs.

Kif5a associates with several members of the HnRNP family, including HnRNPA1 (*Kanai et al., 2004*). HnRNPA1 has been shown to play a role in concentration-dependent stress granule formation in other neurodegenerative diseases including ALS (*Kim et al., 2013*; *Molliex et al., 2015*; *Deshaies et al., 2018*). Therefore, we hypothesized that HnRNPA1 knockdown may be deleterious for survival in the context of optic nerve degeneration. We injected a viral vector expressing either an shRNA construct targeting HnRNPA1 or an shRNA scramble construct 2 weeks prior to optic nerve injury and collected retinas 2 weeks after. HnRNPA1 knockdown significantly decreased RGC survival after injury (*Figure 5i–j*). Together, these data further demonstrate the importance of Kif5a and its cargoes in regulation of RGC survival after neurodegenerative injury.

## Kif5a knockout leads to progressive and dose-dependent RGC degeneration

As Kif5a knockout disrupts protein and perhaps mitochondrial transport in the optic nerve, we tested whether the loss of Kif5a affected RGC survival in the absence of optic nerve injury. Using either *Kif5a* or *Kif5b* floxed adult mice and intravitreal AAV-cre delivery, we quantified RGC survival using Brn3a, a nuclear RGC-specific marker, in retinal wholemounts (*Figure 6a*). RGC density did not change 1 month after Kif5b homozygous knockout (*Figure 6—figure supplement 1*), but we detected a significant 44% decrease in RGC survival 1 month after Kif5a homozygous knockout. This Kif5a-dependent RGC death also led to significant optic nerve atrophy, confirming axonal degeneration in addition to RGC death (*Figure 6b*). We assayed gene dose dependency using a heterozygous *Kif5a^{fl/+}* mouse line with AAV-cre and found a significant decrease of 22% RGC survival, about half of the cell death seen in the homozygous Kif5a knockout (*Figure 6c*; 1-2). To determine if only 44% of RGCs are susceptible to cell death or if Kif5a loss results in progressive degeneration, we aged a cohort of mice to 4 months after knockout and observed an almost complete loss of RGCs (*Figure 6c*; 3-4). To test whether this effect could be a downregulation of Brn3a, we also measured pattern electroretinograms for function of RGCs. Visual responses were almost completely lost 6 months after knockout, quantified by P50 to N95 amplitude (*Figure 6d*). These data indicate that the loss of a specific motor protein Kif5a, but not Kif5b, induces a progressive neurodegeneration as well as a dose-dependent neurodegeneration of RGCs.

We next asked how overexpression of Kif5a would affect RGCs after injury. After validating that our overexpression construct could transport down the optic nerve and bind TUBB3 (*Figure 7a*), we measured the effects on RGC survival. Interestingly, while there were no changes in cell death 1 month post viral injection without injury (*Figure 7b*), there was a 33% decrease in RGC survival after optic nerve crush with Kif5a overexpression compared to control (*Figure 7c–d*). Therefore, maintaining physiological levels of Kif5a transport—and not above or below this level—is crucial for RGC survival after injury.

## Discussion

In diseases such as ALS and glaucoma, axon transport failure has long been proposed to contribute to the pathophysiology of neuronal cell death (*Anderson and Hendrickson, 1974*; *Minckler et al., 1977*; *Quigley and Anderson, 1977*). To increase our understanding of the degenerative process, here we quantify the axon transportome separated from surrounding optic nerve glial and other

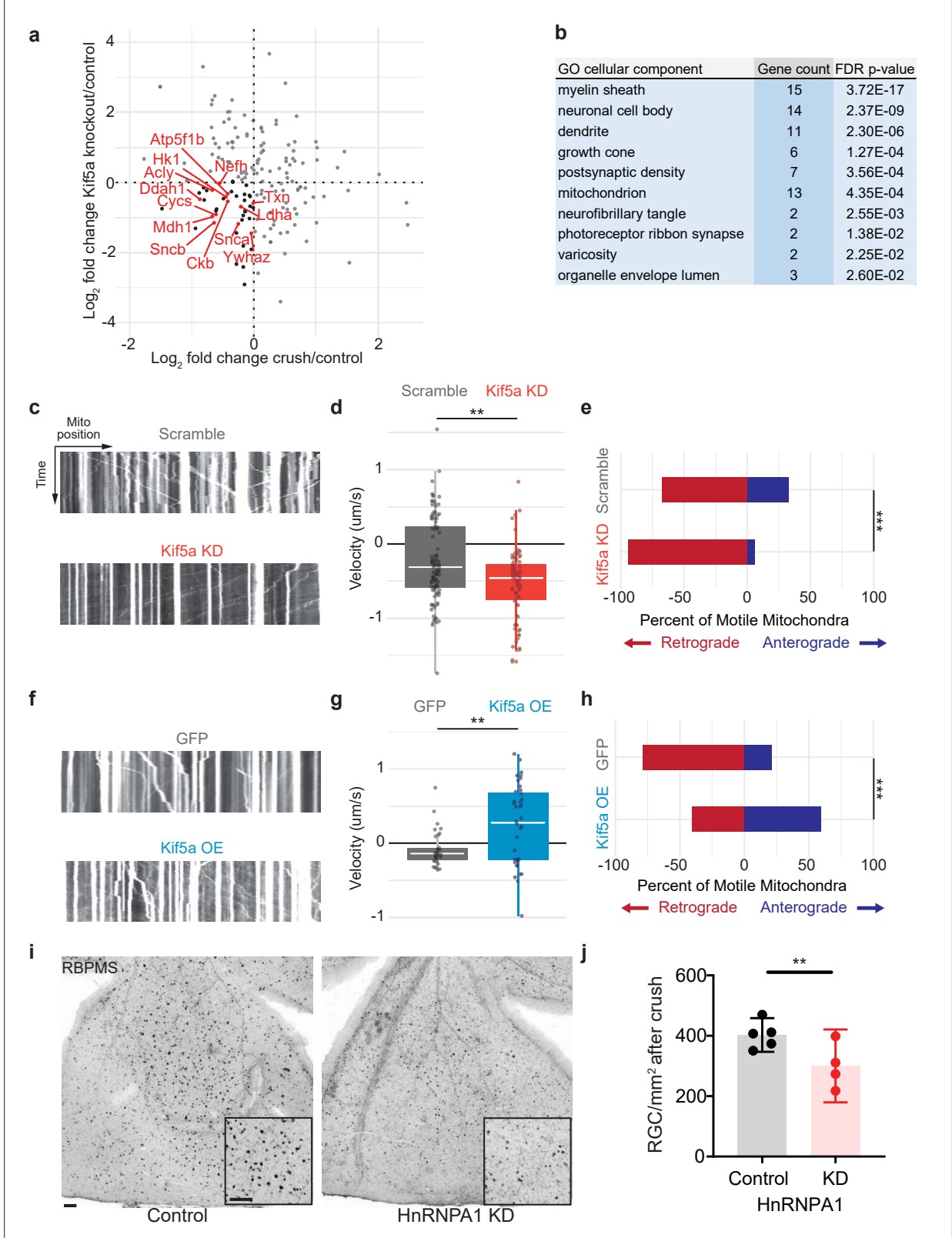

**Figure 5.** Mitochondrial transport is regulated by Kif5a in retinal ganglion cells (RGCs). (**a**) Log$_2$ comparison of protein transport changes in Kif5a knockout (KO) compared to optic nerve crush. Proteins in the lower left quadrant are decreased in both conditions. Proteins in red are identified as associated with Gene Ontology (GO) cellular component 'mitochondrion'. Proteins in black are non-mitochondrial. (**b**) Top GO cellular component ontologies for over-representation with shared hierarchy terms removed. 'Gene count' refers to number of genes in bottom left quadrant (decreased in

*Figure 5 continued on next page*

*Figure 5 continued*

transport after Kif5a KO and optic nerve crush [ONC]) out of total genes in common between the two datasets. Full enrichment list found in **Figure 5— source data 2**. (**c,f**) Representative kymographs of mitochondrial movement in isolated E18 RGC neurites using MitoTracker deep red after viral delivery of either Kif5a shRNA or scramble shRNA (**c**) or Kif5a overexpression or GFP (**f**). Images captured by time-lapse confocal microscopy. (**d,g**) Quantification of average mitochondrial velocity. Each data point represents a single mitochondrion. Boxplot covers the interquartile range with median noted in white. Two-sample, two-tailed t-test, p < 0.0001 (cultures = 3; Kif5a KD mito = 100; neurons = 19, scramble mito = 117; neurons = 18), p = 0.002 (cultures = 3; Kif5a OE mito = 52; neurons = 13, GFP = 42; neurons = 22). (**e,h**) Proportion of retrogradely moving mitochondria versus anterogradely moving mitochondria. Fisher's exact test, p < 0.0001 (top), p = 0.0003 (bottom). (**i**) Representative example of wholemount retinas 2 weeks after ONC stained with RGC-specific marker RBPMS, after either intravitreal adeno-associated virus (AAV)-scramble-shRNA or AAV-HnRNPA1-shRNA injection. For both wholemount and inset: scale bar, 100 µm. (**j**) Quantification of RBPMS + cell density after ONC across entire retinal surface. Each point represents one retina (control n = 5, KD n = 4). Two-sample, two-tailed t-tests, p = 0.039.

The online version of this article includes the following source data for figure 5:

**Source data 1.** Protein-by-protein comparison of fold change in ONC compared to Kif5a knockout from **Figure 5a**.

**Source data 2.** Cellular component Gene Ontology using the PANTHER platform for **Figure 5b**.

cells' proteomes, and for the first time describe specific changes in the complement of transported proteins after injury. Previous approaches to characterizing axon-specific protein responses to injury depended on physically extracting cytoplasm from axons in whole nerve preparations (**Michaelevski et al., 2010**), a technically challenging approach with inconsistent results noted by those authors. We previously established an in vivo protein biotinylation, mass spectrometry-based axon transportome approach which requires amine reactivity to study steady-state protein transport in the visual system. While this labeling method requires available amine-groups, previous work characterizing the types of proteins identified by NHS-biotin did not show any particular bias compared to unlabeled proteomics (**Schiapparelli et al., 2014**; **Schiapparelli et al., 2019**). Here, adapting this technique to NHS-biotin injections within a 24 hr window allowed us to capture spatial and temporal dynamics in axonal protein transport immediately following axon injury and contribute three major new findings.

First, this improved temporal resolution quantitatively detected loss of specific proteins' transport before total protein changes are apparent, as seen comparing western blotting of whole tissue to IP of biotinylated proteins. In fact, many of the axonal changes that occur after injury may begin earlier than previously thought, with rapid changes to the axon transportome. Additional examination of the time course of axon transport changes in the period after injury or in other models of neurodegenerative diseases may now be addressed through such time-resolved quantitative proteomics in future studies. The tradeoff from increasing temporal resolution and dissecting the fraction of transported proteins from the surrounding milieu is a decrease in the total number of identified, labeled peptides available for quantification. Current, state-of-the-art mass spectrometers allowed discovery of some of the transportome changes, but not of transported proteins below the detection threshold. Future advances in proteomics will allow for deeper sequencing and likely to identification of additional biologically relevant transportomic changes.

Second, using this novel approach for quantifying axonal protein transport, we identify a specific transport failure of Kif5a in RGC axons early in the degenerative course following optic nerve injury. Disruptions to the kinesin motor protein family manifest in a variety of neurological diseases. Charcot-Marie-Tooth disease type 2A can be caused by mutations in Kif1b (**Zhao et al., 2001**), congenital fibrosis of the extraocular muscles can be caused by mutations in Kif21a, and a loss-of-function mutation in Kif5a motor domain can result in hereditary spastic paraplegia (**Reid et al., 2002**; **Xia et al., 2003**; **Tessa et al., 2008**; **Morfini et al., 2009**). Different loss-of-function mutations in Kif5a, all affecting the cargo binding domain, are causative in some cases of ALS (**Brenner et al., 2018**; **Nicolas et al., 2018**). In this study, we identified both acute transport failure of Kif5a, without a concomitant change in retinal abundance, after acute optic nerve injury and a decrease in new protein synthesis of Kif5a 3 weeks after initiation of a chronic glaucoma model. These orthogonal experiments using separate protein labeling paradigms both point toward downregulation of Kif5a as part of an overall maladaptive response of RGCs to injury.

Third, these data open a new direction for study of transport biology, to better understand motor protein cargo specificity and physiology. The finding of Kif5a transport failure after injury is especially interesting given the relative lack of change in other motor proteins, such as Kif5b. Increasing evidence has shown that kinesin subtypes and adaptor proteins have at least partial cargo specificity

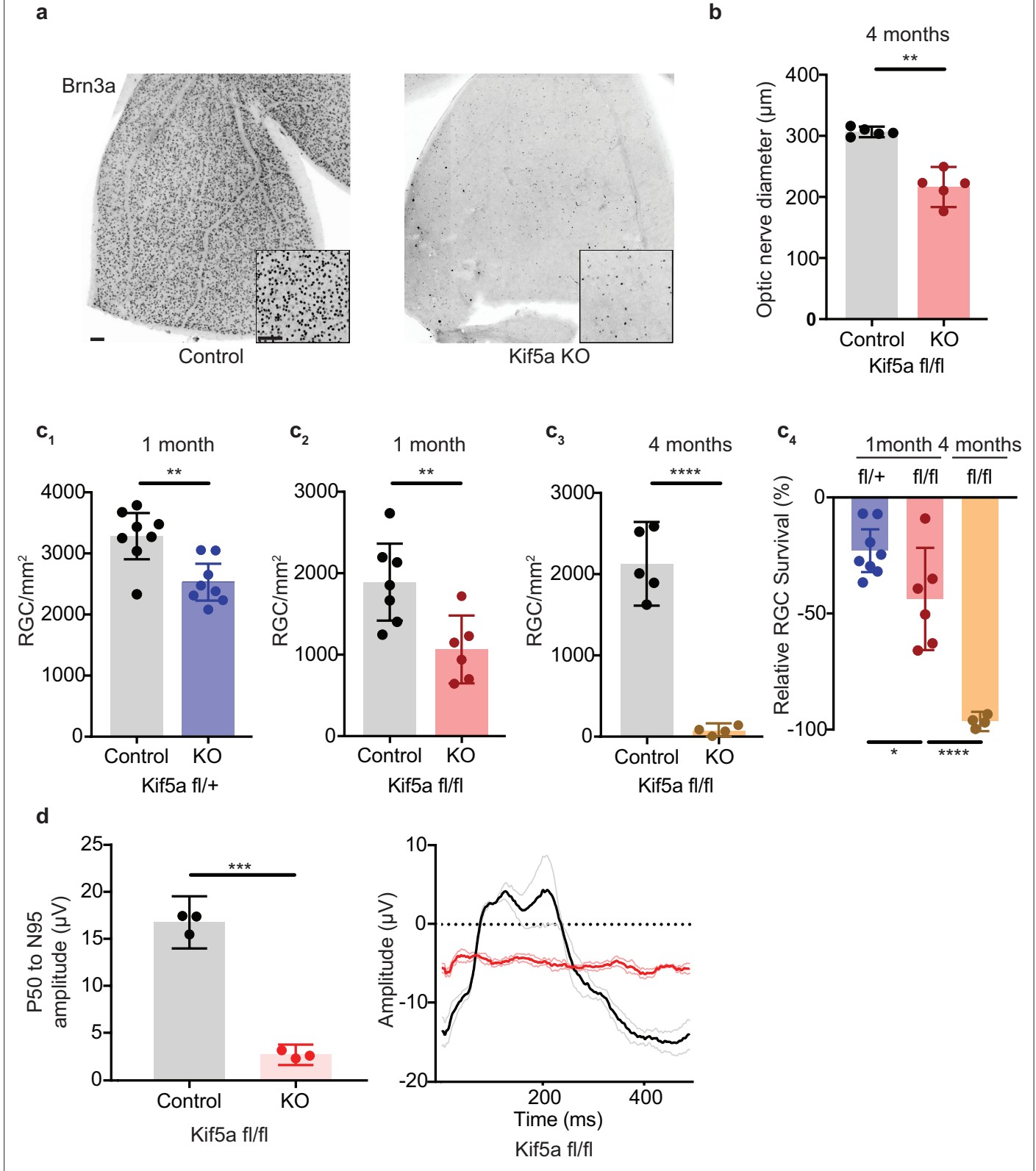

**Figure 6.** Kif5a knockout (KO) leads to progressive, dose-dependent retinal ganglion cell (RGC) degeneration. (**a**) Representative example of wholemount Kif5a$^{fl/fl}$ retinas stained with RGC-specific marker Brn3a, 4 months after either intravitreal adeno-associated virus (AAV)-GFP or AAV-Cre-GFP. Scale bar = 100 µm for both wholemount images and insets. (**b**) Quantification of the average diameter of optic nerve cross-section taken halfway between the optic nerve chiasm and optic nerve head in either control or Kif5a knockout (KO) animals (control n = 5, KO n = 5). Two-sample, two-tailed

*Figure 6 continued on next page*

*Figure 6 continued*

t-test, p = 0.001. (**c**) Quantification of Brn3a + cell density across entire wholemount retinal surface. Each point represents one retina. The graphs, from left to right, show pairwise comparisons between heterozygous Kif5a KO at 1 month after viral injection (control n = 8, KO n = 8), homozygous Kif5a KO at 1 month after viral injection (control n = 7, KO n = 6), homozygous Kif5a KO at 4 months after viral injection (control n = 5, KO n = 4), and a comparison of the three previous experiments normalized to their control population. $C_1$-$C_3$ are two-sample, two-tailed t-tests, p = 0.003, p = 0.008, and p < 0.0001, respectively. $C_4$ is a one-way ANOVA with post hoc Sidak correction. Between the blue and red columns, p = 0.034. Between red and gold columns, p < 0.0001. The bar heights represent the mean, and the error bars represent the 95% confidence interval. (**d**) Pattern electroretinogram quantification from mice 6 months after Kif5a KO or control. On the left, the amplitude difference between the P50 peak and the N95 trough was quantified and compared with a two-sample, two-sided t-test (n = 3), p < 0.0001. The bar heights represent the mean, and the error bars represent the 95% confidence interval. On the right, the average traces of each condition are bold, with lighter shades ± SEM.

The online version of this article includes the following figure supplement(s) for figure 6:

**Figure supplement 1.** Loss of Kif5b does not lead to RGC cell death.

(***Chevalier-Larsen and Holzbaur, 2006***), yet the extent of shared and specific cargo between kinesin proteins is not fully understood. Through IP and immunoisolation assays, several synaptic proteins such as Snap25, synaptotagmin, and syntaxin 1a have been identified as cargoes of the Kif5 family (***Toda et al., 2008***), but only a few proteins have been shown to be specific to Kif5a, such as GABARAP (***Nakajima et al., 2012***). Quantitative changes in cargo transport after the loss of a specific motor protein may give insight into the role of that motor, and future studies could expand on these data by directly comparing the axon transportome of different kinesin isoforms.

Specificity of function between these closely related motor proteins was also an important finding in these data. Kif5a loss, but not Kif5b loss, resulted in the progressive, dose-dependent degeneration of adult RGCs; this specificity mirrored the specificity of Kif5a's decreased axon transport detected after optic nerve injury, and the enrichment of *Kif5a* in mouse and human RGCs contrasted with the ubiquitous expression of *Kif5b*. Although overexpression of Kif5a did not increase RGC survival after injury, the conservation of RGC-enriched *Kif5a* expression across species suggests a strategy of manipulating Kif5a therapeutically to restore axonal transport after injury or in neurodegenerative diseases like glaucoma. This approach, along with further characterization of Kif5 expression in RGC subtypes, would be intriguing next steps in understanding motor protein biology.

Although here we describe initial data on the axon transportome cargoes dependent on Kif5a for their RGC axon localization, we do not yet understand which cargoes of Kif5a contribute to optic neuropathy, especially in vivo. By quantifying the transportomic changes in an adult acute knockout model, it is possible that we identified only the subset of proteins whose transport was not adequately compensated by other mechanisms or motors. Indeed, bioinformatic analysis indicates that many of these protein cargoes have known interactions with each other, and several have previously been reported as associated with degenerative diseases in other systems. Another possibility is that these proteins, instead of being direct cargoes of Kif5a, are downregulated or mis-localized after Kif5a loss. Increased protein degradation would also contribute to decreased detection of biotinylated proteins, as detection of transported proteins is the net of protein transport and protein degradation. Finally, these data are limited to transported proteins. Kif5a can act as a motor for fast axonal transport of membranous organelles, slow axonal transport of proteins, and transport of neurofilaments (***Xia et al., 2003***). In addition, cargoes can include mRNA and related P-bodies and stress granules for local protein synthesis (***Loschi et al., 2009***; ***Oh et al., 2013***). Our finding that knockdown of one such ribonucleoprotein cargo of Kif5a, HnRNPA1, results in exacerbated degeneration after injury suggests that the loss of Kif5a-dependent RNA and other organelle transport may also contribute to neurodegeneration. In particular, stress granules have been implicated in at least one form of glaucoma (***Lachke et al., 2011***), which combined with our data suggests exploring the role of both stress granules and mitochondria as disease-relevant Kif5a cargo as an exciting future direction.

Indeed, the over-representation of mitochondrial-associated proteins demonstrating decreased axon transport in both of our datasets point toward a lack of mitochondrial transport after loss of Kif5a. The short time point assayed in our optic nerve injury model (24 hr) is consistent with the rapid halt of mitochondrial transport in RGC axon in an ocular hypertension/glaucoma model (***Takihara et al., 2015***). Kif5 motors regulate mitochondrial trafficking in *Drosophila* and mammalian hippocampal neurons through the adaptor proteins Milton and Miro (***Glater et al., 2006***; ***Macaskill et al., 2009***). Mitochondrial localization in axons regulates both neuronal survival and axon regeneration. In

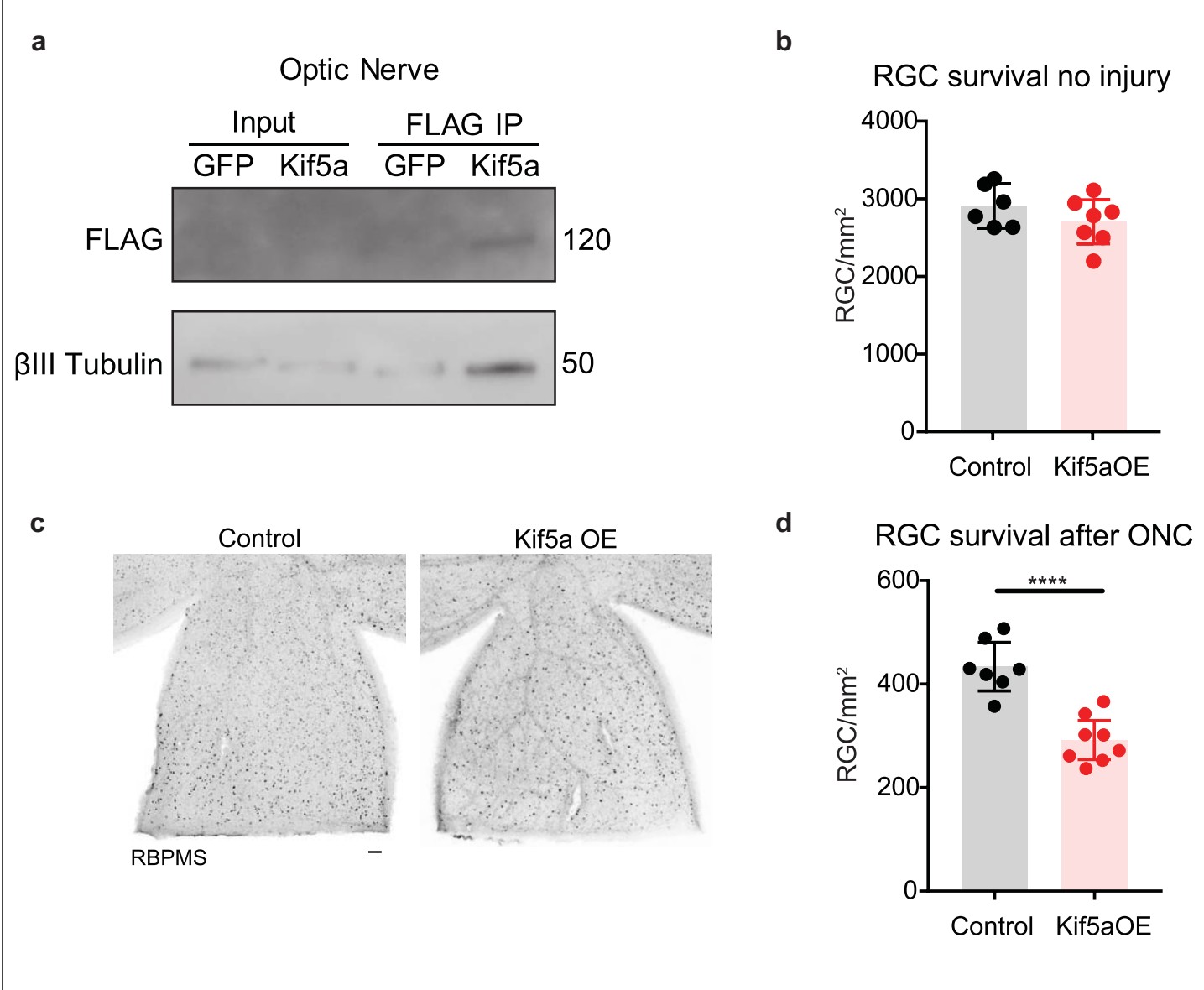

**Figure 7.** Kif5a overexpression exacerbates retinal ganglion cell (RGC) death. (**a**) Optic nerves were collected 2 weeks after intravitreal injection of adeno-associated virus (AAV)-Kif5a-FLAG. Immunoprecipitation showed the presence of FLAG-tagged protein at the molecular weight of Kif5a. Co-immunoprecipitation of β-III tubulin confirmed that overexpressed Kif5a transported to the optic nerve and bound to the cytoskeleton. (**b**) RGC survival was not affected 4 weeks after viral injection of Kif5a compared to a control GFP virus (control n = 6, OE n = 7). Two-tailed, two-sample, t-test. (**c**) Representative example of wholemount retinas stained with RBPMS, an RGC-specific marker, 2 weeks after optic nerve crush, injected with either AAV-GFP or AAV-Kif5a-FLAG 2 weeks before crush. Scale bar, 100 μm. (**d**) Quantification of RBPMS + cell density after optic nerve crush (ONC) across entire retinal surface. Each point represents one retina (control n = 7, OE n = 8). Two-sample, two-tailed t-tests, p < 0.0001.

The online version of this article includes the following source data for figure 7:

**Source data 1.** Raw western blots for *Figure 7*.

*Caenorhabditis elegans*, mitochondria localize to injured axons during the normal course of regeneration (*Han et al., 2016*). In non-regenerating neurons, like mammalian RGCs, manipulating mitochondrial mobilization through overexpression of Armcx1 produces a similar survival and regenerative effect (*Cartoni et al., 2016*). Previous studies have shown that Kif5a may regulate mitochondrial transport in zebrafish (*Campbell et al., 2014*) and cultured hippocampal neurons (*Chen and Sheng, 2013*). However, it was not known if Kif5 motors regulate mitochondrial trafficking in RGC neurons in vitro or in vivo or if manipulating Kif5a alone is sufficient to affect mitochondrial transport, both hypotheses

now supported with these new data. Combining these two findings, we hypothesize that targeted anterograde transport of mitochondria to the injured mammalian axon could potentially enhance the effects already seen with Armcx1-mediated mitochondrial mobilization. Overexpression of Kif5a in RGCs in vitro preferentially increases anterograde mitochondrial transport, increasing mitochondrial concentration in the axon, raising the question of whether overexpressing Kif5a in vivo would promote RGC survival. Interestingly, overexpression of Kif5a led to increased cell death after injury. However, Kif5a abundance in the retina is unchanged after optic nerve injury, suggesting that loss of optic nerve localization is specific to axon transport failure rather than to deficit of Kif5a expression itself. Regulation of kinesin transport is known to depend on phosphorylation states (*Morfini et al., 2001*), calcium-dependent inhibition by mitochondrial cargo (*Wang and Schwarz, 2009*), conformational state changes by kinesin light chains, inhibition by MAP2 at the axon initial segment (*Gumy et al., 2017*), and autoinhibition of the motor domain by inactive kinesin heavy chains (*Cai et al., 2007*). Removing autoinhibitory domains or truncation of the motor beyond the first coiled-coil domain creates constitutively active kinesins, potentially circumventing any inherent axonal inhibition (*Yang et al., 2016*). Future work specifically restoring anterograde protein and mitochondrial trafficking in the face of axonal injury, perhaps using constitutively active Kif5a mutants, could rescue cell death and provide therapeutic avenues to counter the neurodegenerative process.

# Materials and methods

**Key resources table**

| Reagent type (species) or resource | Designation | Source or reference | Identifiers | Additional information |
|---|---|---|---|---|
| Strain, strain background (*Mus musculus*) | Wild-type | The Jackson Laboratory | C57BL/6 | Pooled sexes |
| Strain, strain background (*Mus musculus*) | Kif5a<sup>fl/fl</sup> | David Williams Laboratory | MMRRC:MMRRC_000224-UCD | Pooled sexes |
| Strain, strain background (*Mus musculus*) | Kif5b<sup>fl/fl</sup> | David Williams Laboratory | JAX stock:008637 | Pooled sexes |
| Strain, strain background (*Rattus norvegicus*) | Wild-type | The Jackson Laboratory | Sprague-Dawley | Pooled sexes |
| Cell line (Human) | HEK 293T | ATCC | 293T | |
| Other | Viral Vector AAV2-GFP | UPenn Vector Core | AV-2-PV0101 | |
| Other | Viral Vector AAV2-cre-GFP | UPenn Vector Core | AV-2-PV2004 | |
| Other | Viral Vector AAV2-Kif5a | This paper | VectorBuilder ID: https://en.vectorbuilder.com/vector/VB170708-1017enf.html | Sui Wang Laboratory; Vector Map from VectorBuilder |
| Other | Viral Vector AAV2-Kif5a shRNA | This paper | VectorBuilder ID: https://en.vectorbuilder.com/vector/VB180108-1217eru.html | Sui Wang Laboratory; Vector Map from VectorBuilder |
| Antibody | Rabbit anti-Kif5a | Abcam | Ab5628 | (1:1000) |
| Antibody | Rabbit anti-Kif5c | Abcam | Ab192883 | (1:1000) |
| Antibody | Goat anti-biotin | Thermo | 31852 | (1:1000) |
| Antibody | Rabbit anti-FLAG | CST | 14793S | (1:2000) |
| Antibody | Rabbit anti-B3Tubulin | CST | 5568S | (1:2000) |
| Antibody | Rabbit anti-GAPDH | CST | 2118 | (1:2000) |

*Continued on next page*

*Continued*

| Reagent type (species) or resource | Designation | Source or reference | Identifiers | Additional information |
|---|---|---|---|---|
| Antibody | Rabbit anti-SNCB | Sigma | SAB1100305 | (1:2000) |
| Antibody | Rabbit anti-GAP43 | Novus | NB300 | (1:5000) |
| Antibody | Rabbit anti-ARF3 | Abcam | Ab154383 | (1:5000) |
| Antibody | Goat anti-mouse | Abcam | Ab6789 | (1:2000) |
| Antibody | Goat anti-rabbit | Abcam | Ab6721 | (1:2000) |
| Antibody | Donkey anti-goat | Abcam | Ab97110 | (1:2000) |
| Antibody | Mouse anti-Brn3a | Millipore | MAB1585 | (1:100) |
| Antibody | Rabbit anti-RBPMS | PhosphoSolutions | 1830-RBPMS | (1:300) |

## Animal lines

C57BL/6 mice or Sprague-Dawley rats of either sex were used for all wild-type experiments. *Kif5a$^{fl/fl}$* and *Kif5b$^{fl/fl}$* from a C57Bl/6 background were a gift from David Williams (*Xia et al., 2003*).

## Intravitreal injection

Sprague-Dawley rats or mice (30–45 days of age) were used for all experiments. A microinjector pressure system (Picosprizer II) with a pulled glass micropipette in a micromanipulator was used to inject reagents intravitreally. For viral injections, mice were injected intravitreally with 1.5 µl volume of AAV2-GFP or AAV2-cre-GFP (UPenn Vector Core) (titers ranged from 0.5 to 1 × 10$^{13}$ genome copies/ml). For in vivo biotinylation experiments, 5 mg of NHS-biotin (EZ-Link from Pierce) were dissolved in 300 µl of sterile DMSO immediately prior to use, as described previously (*Schiapparelli et al., 2019*). For in vivo AHA experiments, we injected each eye with ~5 µl of 400 mM AHA in PBS. Briefly, intravitreal injections (5 µl for rats, 1.5 µl for mice) were given to both eyes. The injections were given twice over 24 hr under deep anesthesia with either 0.5 mg/kg medetomidine and 75 mg/kg ketamine intraperitoneal, or 2% isoflurane under nose cone. The eyes were treated with topical antibiotics and analgesics.

## Optic nerve crush

Under deep anesthesia, the optic nerve was exposed and crushed using fine forceps (Dumont #5) at 1.5 mm behind the optic nerve head for 5 s. Care was taken to avoid damaging the blood supply to the retina. Sham surgeries exposed the optic nerve, but no crush was given. All optic nerve crush procedures were performed by a surgeon blinded to the viral treatment. Postoperatively, animals were allowed to recover on a heating pad and were given subcutaneous injections of buprenorphine hydrochloride, 0.1 mg/kg, twice a day for 3 consecutive days to minimize discomfort.

## Circumlimbal glaucoma model

Baseline intraocular pressure (IOP) of adult rats was measured using a rebound tonometer (Tonolab; iCare, Helsinki, Finland) under inhaled isoflurane anesthesia. Eyes were randomized to either circumlimbal glaucoma or sham. To induce glaucoma, a circumlimbal suture (7/0, nylon) was tied around the eye 1.5 mm behind the limbus with five anchor points. In the glaucoma group, the suture was tied tightly, with intraoperative IOP measurements greater than 40 mmHg. In sham eyes, the suture was tied loosely. IOP measurements were taken every other day between 2 PM and 6PM under isoflurane anesthesia.

## Pattern electroretinogram

Anesthetized mice were placed on a feedback-controlled heating pad (TCAT-2LV, Physitemp Instruments Inc, Clifton, NJ) to maintain body temperate at 37°C. Simultaneous binocular PERG recording was completed with the Miami PERG system (Intelligent Hearing Systems, Miami, FL) according to published protocol (*Chou et al., 2018*). Eye drops were added as needed to prevent corneal drying

(Systane Ultra Lubricant Eye Drops, Alcon Laboratories, Ft Worth, TX). The reference electrode was placed subcutaneously on the back of the head between the two ears and the ground electrode was placed at the root of the tail. The signal electrode was placed subcutaneously on the snout for the simultaneous acquisition of left and right eye responses. Two 14 cm × 14 cm LED-based screens were placed 10 cm in front of each eye. The pattern remained at a contrast of 85% and a luminance of 800 cd/m$^2$, and consisted of four cycles of black-gray elements, with a spatial frequency of 0.052 c/d. Upon stimulation, the PERG signals were recorded by asynchronous binocular acquisition. Two consecutive recordings of 200 traces were averaged to achieve one readout. The first positive peak in the waveform was designated as P50 and the second negative peak as N95. The investigators who measured the amplitudes were blinded to the treatment of the samples.

## Immunohistochemistry

At different time points after surgeries, animals were deeply anesthetized and transcardially perfused with 4% PFA in PBS. Optic nerves and retinas were dissected and fixed in 4% PFA for 1 hr and subsequently washed in PBS. Optic nerves were sequentially incubated in 15% and 30% sucrose at 4°C overnight before mounting in optimal cutting temperature mounting medium for sectioning. Retinas were kept in PBS at 4°C until ready for antibody staining. Retinas were incubated with mouse anti-Brn3a (1:100; Millipore, MAB1585) and rabbit anti-Rbpms (1:300; PhosphoSolutions, 1830-RBPMS), both specific markers of RGCs, and anti-Kif5a (1:300, Abcam ab5628). Secondary antibodies were Alexa Fluor 488-, 594-, or 647-conjugated, highly cross-adsorbed antibodies (1:500; Invitrogen). Wholemount retinal images were acquired using a laser scanning confocal microscope (Carl Zeiss 880 or Olympus FV500) and quantified by a researcher blinded to the experimental condition. Biotinylated optic nerves sections were incubated with 100 mM glycine in PBS for 2 hr and endogenous peroxidase activity was blocked with 0.5% of $H_2O_2$ and 1% normal goat serum (NGS) in PBS for ½ hr. Sections were blocked with 10% NGS in PBS for 1 hr and incubated overnight with ABC reagent (1 drop of A and 1 drop of B in 5 ml PBS with 1% NGS, Vector Lab). Signal was amplified using tyramide amplification system (TSA kit, PerkinElmer) conjugated to Alexa Fluor 488.

## RGC quantification

The RBPMS quantification were performed in a masked fashion as previously described (*Wang et al., 2015*). Briefly, the retinas were divided into four quadrants, and one digital micrograph was taken from a fixed distance from the periphery of each of the four fields. Whole retina Brn3a was quantified using a semi-automated protocol in Velocity (PerkinElmer) and ImageJ in a masked fashion. Tiled images of entire retinal wholemounts were imported into ImageJ for quantification of total area. During this step, care was taken to remove sections of retina damaged during tissue processing. These modified images were then moved to Velocity, where 'Find object' using thresholds for minimum object size, intensity, and touching neighbors. Once the protocol was set for one randomly selected retina, it was applied to all retinas in the experiment.

## FUNCAT

NHS-Azide-injected samples were processed for click chemistry according to the following protocol, modified from previous studies (*Dieterich et al., 2010*; *Hinz et al., 2012*). Sections or wholemounts were transferred to Eppendorf tubes or six-well plates with reaction mixture composed of 100 μm tris[(1-benzyl-1H-1,2,3-triazol-4-yl)methyl]amine (TBTA, Sigma) dissolved in 4:1 tBuOH/DMSO (Sigma), 100 μm $CuSO_4$ (Sigma), 1.25 μm Alexa Fluor 488 alkyne (Invitrogen), and 250 μm tris(2-carboxyethyl) phosphine (TCEP, Sigma). The reaction proceeded overnight at room temperature.

## Biotin click reaction

Optic nerves or retinas were lysed in 0.5% SDS in PBS plus a cocktail of endogenous protease inhibitors (Complete Protease Inhibitor Cocktail Tablets, Roche) by homogenizing and sonicating with 10 pulses using a tip sonicator (Sonic Dismembrator model 100, Fisher Scientific). Samples were boiled for 10 min and cooled to room temperature. Any remaining insoluble material was resuspended with additional sonication pulses. Protein concentration in the suspension was measured, and aliquots of 1.5 mg of protein suspension were transferred to eppendorf tubes. NHS-azide or AHA that was incorporated into proteins was labeled with PEG4 carboxamide-Propargyl Biotin (biotin-alkyne) (Invitrogen)

by click chemistry reaction performed in the total protein suspension as described previously (*Speers and Cravatt, 2009*; *Hulce et al., 2013*). For AHA samples, click reactions were done using biotin-alkyne labeled with heavy stable carbon and nitrogen isotopes: biotin-beta-alanine-$^{13}C_3$,$^{15}N$-alkyne [Biotin Propargyl amide] or light isotope forms of the alkyne (#7,884 and #7889, respectively; Setareh Biotech) in the different experimental groups. Centrifugation steps that can result in loss of NHS azide- or AHA-labeled material were avoided. For each reaction, an aliquot of 1.5 mg of protein suspension was used, adding PBS to reach 346 µl before adding the click reaction reagents. We added the following reagents in sequence, vigorously vortexing after each addition: 30 µl of 1.7 mM TBTA (Sigma) dissolved in 4:1 *tert*-butanol/DMSO (Sigma), 8 µl of 50 mM $CuSO_4$ dissolved in ultrapure water (Sigma), 8 µl of 5 mM of PEG4 carboxamide-propargyl biotin (biotin-alkyne) (Invitrogen) dissolved in DMSO, and 8 µl of 50 mM TCEP (Sigma) dissolved in water. The click reactions were incubated at room temperature for 1–2 hr or overnight with gentle rotation at 4°C. After the completion of each click reaction, samples were aliquoted by transferring 200 µl of each click reaction suspension to 2 ml eppendorf tubes.

## Fresh tissue collection

Rats or mice were euthanized with $CO_2$ and decapitated for brain removal. The tissue was frozen immediately in isopentane in dry ice and stored at –80°C for biochemistry studies. Co-immunoprecipitation studies continued directly without freezing.

## IP and western blot

For western blot, after defrosting frozen tissue the retinas or optic nerves were dissected and homogenized by sonication in 50–100 µl RIPA Lysis Buffer with protease and phosphatase inhibitor cocktail (Holt, 78440). Samples were centrifuged at 10,000× *g* for 10 min at 4°C, and the supernatants collected. Total protein concentration was determined using a BCA assay (Pierce BCA Protein Assay Kit 23225).

To purify biotinylated proteins for western blot validation, 0.8–2 mg of proteins from ON or retinal homogenates in a total volume of 1 ml were incubated with 30 µl neutravidin beads (Thermo) at 4°C overnight. The beads were then washed 6 times with 1 ml of RIPA buffer. The bound biotinylated proteins where eluted from the beads with 50 µl Laemmli sample buffer containing 2.5% of 2-mercaptoethanol.

For co-immunoprecipitation, retinas and optic nerves were mechanically homogenized in buffer (20 mmol/l HEPES, pH 7.4, 150 mmol/l NaCl, 5 mmol/l EDTA, 0.5% Triton, 50 mmol/l NaF, 1 mmol/l sodium orthovanadate, 1 mmol/l DTT, and protease inhibitors). After centrifugation at 10,000× *g* for 10 min at 4°C, the clarified extracts were used for IP using appropriate antibodies (2 µg rabbit anti-FLAG, CST, 14793S) and 20 µl protein G sepharose (Millipore, Fastflow) for 3 hr to at 4°C. The beads were washed 3–5 times with lysis buffer, and the immunoprecipitated proteins were eluted with 2× Laemmli buffer for western blotting.

For all electrophoresis, 10–30 µg of protein was loaded on a 4–20% Tris-Glycine Gel (Thermo Fisher Scientific XP04202) and transferred to an activated 0.2 µm PVDF membrane for immunoblotting. The membranes were saturated with Ponceau S stain for 5 min and imaged, before washing with ddH$_2$O. The membranes were saturated with TBS 1×, 0.05% Tween-20, and 5% nonfat dry milk for 1 hr at room temperature, then incubated overnight at 4°C with rabbit anti-Kif5a (1:1000, Abcam, ab5628), rabbit anti-Kif5c (1:1000, Abcam, ab192883), goat anti-biotin (1:1000, Thermo, 31852), rabbit anti-FLAG (1:2000, CST, 14,793S), rabbit anti-β-III tubulin (1:2000, 5568S, CST), rabbit anti-GAPDH (1:2000, CST, 2118), rabbit anti-SNCB (1:1000, Sigma), rabbit anti-GAP43 (1:5000, Novus), rabbit-ARF3 (1:1000, Abcam). All membranes were washed with TBS-Tween between incubations. The membranes were then saturated with peroxidase-conjugated goat anti-mouse, anti-rabbit, or anti-goat (1:2000, Abcam, ab6789, ab6721, and ab97110) secondary antibodies for 3 hr at room temperature and revealed with SuperSignal West Femto Chemiluminescent Substrate using ImageQuant LAS4000. Stripping to ensure equal loading was done with ReBlot Plus Strong Antibody Stripping Solution (EMD Millipore 2504). All quantifications were done with ImageJ densitometry analysis.

## In vitro cell culture

HEK 293T cells were grown to 100% confluence in 75 or 150 cm$^2$ flasks, dissociated, and resuspended with TrypLE, transferred to 15 ml falcon tubes, centrifuged for 5 min at 1000 rpm at room temperature, and washed three times in Dulbecco's modified PBS (DPBS, Gibco). Cells were incubated in suspension with 1 mg/ml of biotin-beta-alanine in 10 ml of DPBS at 4°C with gentle rotation. Cells were washed three times in DPBS, pelleted by centrifugation, and frozen on dry ice. The biotinylated cell pellets were homogenized in RIPA buffer containing 1% NP40, 0.5% sodium deoxycholate, 0.1% SDS, 150 mM NaCl, 1 mM EDTA, and 25 mM TrisHCl, pH 7.4. Lysates were rotated at 4°C for 30 min and centrifuged at 10,000 $g$ for 10 min at 4°C to remove DNA and cell debris. After measuring the protein concentration using the DC Protein Assay Kit II (Bio-Rad), the lysates were aliquoted by transferring 1–2 mg of protein to 2 ml eppendorf tubes.

## Mass spectrometry

Soluble peptides were pressure-loaded onto a 250 μm i.d. capillary with a kasil frit containing 2 cm of 10 μm Jupiter C18-A material (Phenomenex) followed by 2 cm, 5 μm Partisphere strong cation exchanger (Whatman). This column was washed with Buffer A after loading. A 100 μm i.d. capillary with a 5 μm pulled tip packed with 15 cm, 4 μm Jupiter C$_{18}$ material (Phenomenex) was attached to the loading column with a union and the entire split-column (loading column–union–analytical column) was placed in line with an Agilent 1100 quaternary HPLC (Palo Alto, Santa Clara, CA). For transportome analysis, the sample was analyzed using a modified four-step separation described previously (*Washburn et al., 2001*). The buffer solutions used were 5% acetonitrile/0.1% formic acid (Buffer A), 80% acetonitrile/0.1% formic acid (Buffer B), and 500 mM ammonium acetate/5% acetonitrile/0.1% formic acid (Buffer C). Step 1 consisted of a 35 min gradient from 0% to 55% Buffer B, a 5 min gradient from 55% to 70% Buffer B, 10 min 100% Buffer B and 27 min 100% Buffer A. Steps 2–3 had the following profile: 5 min of x% Buffer C with (100×)% buffer A, a 10 min Buffer A, a 5 min gradient from 0% to 15% Buffer B, a 70 min gradient from 15% to 55% Buffer B, a 5 min gradient from 55% to 100% Buffer B, a 5 min 100% Buffer B, and 20 min 100% Buffer A. The Buffer C percentages (X) were 10 and 40 for the steps 2–3, respectively. In the last step, the gradient contained: 5 min of 90% Buffer C with 10% Buffer B, a 10 min Buffer A, a 5 min gradient from 0% to 15% Buffer B, a 70 min gradient from 15% to 55% Buffer B, a 5 min gradient from 55% to 100% Buffer B, a 5 min 100% Buffer B, and 20 min 100% Buffer A. For whole proteome analysis, the sample was analyzed using a 11-step separation exactly described as above except the Buffer C percentages (X) were 10%, 15%, 20%, 30%, 40%, 50%, 60%, and 80% for the steps 2–9, respectively. In the last two steps (i.e. 10 and 11), the gradient contained: 1 min 100% Buffer A, 5 min of 100% Buffer C, a 5 min Buffer A, a 5 min gradient from 0% to 15% Buffer B, 70 min gradient from 15% to 55% Buffer B, 5 min gradient from 0% to 10% Buffer B, 75 min gradient from 10% to 45% Buffer B, 10 min 100% Buffer B, and 10 min 100% Buffer A. As peptides eluted from the microcapillary column, they were electrosprayed directly into a Velos mass spectrometer (Thermo Fisher) with the application of a distal 2.4 kV spray voltage. A cycle of one full-scan FT mass spectrum (300–2000 m/z) at 60,000 resolution followed by 20 data-dependent IT MS/MS spectra at a 35% normalized collision energy was repeated continuously throughout each step of the multidimensional separation. Application of mass spectrometer scan functions and HPLC solvent gradients were controlled by the Xcalibur data system.

## MS data analysis

MS2 (tandem mass spectra) was extracted from the XCalibur data system format (.RAW) into MS1 and MS2 formats using in-house software (RAW_Xtractor) (*McDonald et al., 2004*). MS2 remaining after filtering were searched with Prolucid (*Xu et al., 2015*) against the UniProt_rat_03-25-2014 or UniProt_mouse_20170219_04-17-2017 concatenated to a decoy database in which the sequence for each entry in the original database was reversed (*Cociorva et al., 2007*; *Peng et al., 2002*). All searches were parallelized and performed on a Beowulf computer cluster consisting of 100 1.2 GHz Athlon CPUs (*Sadygov et al., 2002*). No enzyme specificity was considered for any search. For NHS-biotin samples, the following modifications were searched for analysis for transportome analyses: a static modification of 57.02146 on cysteine for all analyses, a differential modification of 226.0776 on lysine for modified peptides. For AHA samples, the following modifications were searched: a static modification of 57.02146 on cysteine for all analyses, a differential modification of 523.2749 on methionine

for AHA to detect the standard biotin-alkyne, and a differential modification of 351.1774 (heavy) and 347.1702 (light) on methionine to detect the different AHA isotope biotin-alkynes. Prolucid results were assembled and filtered using the DTASelect (version 2.0) program (*Tabb et al., 2002*; *Cociorva et al., 2007*). DTASelect 2.0 uses a linear discriminant analysis to dynamically set XCorr and DeltaCN thresholds for the entire dataset to achieve a user-specified false discovery rate (FDR). In DTASelect, the modified peptides were required to be partially tryptic, less than 10 ppm deviation from peptide match, and an FDR at the protein level of 0.01. The FDRs are estimated by the program from the number and quality of spectral matches to the decoy database. For all datasets, the protein FDR was <1% and the peptide FDR was <1%.

The Datasets from 'Control Rat Transportome', 'Optic Nerve Crush Rat Transportome', 'Control Mouse Transportome', and 'Kif5a KO Mouse Transportome' are composed of the total detected biotinylated proteins in 3, 3, 2, and 3 individual MS runs, respectively. Each sample was prepared by pooling tissue from 8 to 10 rats or 18 to 20 mice. NSAF was exported per UniProt accession ID for further quantitative analysis. Quantification was done using label-free analysis comparing spectral count in the different experimental groups with the Identification_STAT_COMPARE tool from IP2 software.

## Human and mouse retina dissociation and scRNA-seq

We analyzed a subset of recently published data (*Getter et al., 2019*; *Hoang et al., 2020*; *Lu et al., 2020*) which may be referenced for a more complete methodology.

Briefly, for tissue dissection and cell dissociation, a human globe from an 86-year-old Caucasian female who died of a myocardial infarction and had no known ocular disease other than cataracts was obtained from the Alabama Eye Bank (Birmingham, AL) and processed within 3.3 hr after death. The study was approved by the Johns Hopkins Institutional Review Board. The neural retina and RPE/choroid were dissected from the globe in ice-cold PBS. First, a circular incision was made on the sclera, behind the limbus, to remove the anterior parts, lens, and vitreous body. The neural retina was then peeled off from the eyecup and dissociated using the Papain Dissociation System (Worthington) following the manufacturer's instructions. RPE cells were dissociated from the eyecup by incubating with 2 ml of 0.05% EDTA (Thermo Fisher Scientific) for 20 min at 37°C. Dissociated cells were resuspended in ice-cold PBS, 0.04% BSA, and 0.5 units/μl RNase inhibitors.

CD1 mice between 7 and 9 weeks of age were purchased from Charles River Laboratories (Wilmington, MA). All experimental procedures were pre-approved by the institutional animal care and use committee of the Johns Hopkins University School of Medicine. Mice were euthanized, and eye globes were removed and incubated in ice-cold PBS. Retinas were dissected, and cells were dissociated using the Papain Dissociation System. In total, four biological replicates were used for the mouse scRNA study. Each replicate contained four retinas from each of two male and two female mice. Dissociated cells were resuspended in ice-cold PBS, 0.04% BSA, and 0.5 units/μl RNase inhibitors. Cell count and viability were assessed by trypan blue staining.

For scRNA-seq, dissociated cells (~10,000) were loaded into a 10× Genomics Chromium Single Cell system (10× Genomics, CA) using v2 chemistry following the manufacturer's instructions (*Zheng et al., 2017*). Libraries were pooled and sequenced on Illumina NextSeq with ~200 million reads per library. Sequencing results were processed through the Cell Ranger 2.1.1 pipeline (10× Genomics) with default parameters. Seurat version 2.3.1 (*Butler et al., 2018*) was used to perform downstream analysis following the standard pipeline using cells with more than 200 genes and 1000 UMI counts, resulting in 16,659 mouse cells and 14,286 human cells. Samples were aggregated, and cell clusters were annotated based on previous literature (*Kiser et al., 2019*). A t-distributed stochastic neighbor-embedding dimension reduction was performed on the top principal components learned from high variance genes. Mclust version 5.4 was used to cluster cells in the t-distributed stochastic neighbor-embedding space, at which point cell type identity of clusters was assigned based on the expression of known marker genes for either retinal or nonretinal tissue.

## Mitochondrial tracking in RGCs

RGCs were purified from litters of E18 rats or mice as described previously (*Moore et al., 2009*). RGCs were seeded in low density as 2000 cells/cm$^2$ on μ-dish 35 mm (Ibidi) in definite media: Neurobasal (Life Technologies) supplemented with penicillin-streptomycin (100 U/ml; Sigma-Aldrich Corp.), insulin

(5 µg/ml; Sigma-Aldrich Corp.), sodium pyruvate (1 mM; Sigma-Aldrich Corp.), L-glutamine (1 mM; Sigma-Aldrich Corp.), triiodothyronine (T3; 40 ng/ml; Sigma-Aldrich Corp.), *N*-acetyl cysteine (5 mg/ml; Sigma-Aldrich Corp.), B-27 Supplement (Gibco), Sato Supplement(1 µg/ml transferrin, 1 µg/ml BSA, 2 nM progesterone, 0.16 µg/ml putrescine, and 0.4 ng/ml sodium selenite), brain-derived neurotrophic factor (BDNF; 50 ng/ml; Peprotech, Rocky Hill, NJ), CTNF (10 ng/ml; Peprotech), forskolin (5 nM; Sigma-Aldrich Corp.), and basic fibroblast growth factor (10 ng/ml; Peprotech). After 24 hr, Kif5a or anti-Kif5a shRNA were transduced by AAV vector into the cultured RGCs (gift of Dr Sui Wang). After 5 days on the dish, mitochondria were labeled with 50 nM of MitoTracker Deep Red FM (Invitrogen) for 15 min and imaged every 1.5 s for 3 min by confocal microscope with incubation (LSM880, Zeiss). From the time-lapse images, we generated kymograph using ImageJ 'Multiple Kymograph' plug-in for ImageJ submitted by J Rietdorf and A Seitz (European Molecular Biology Laboratory, Heidelberg, Germany) to categorize mitochondrial state (stopped, fluttering, anterograde, and retrograde transport) and calculate speed of the transport as described previously (*Yokota et al., 2015*).

### Protein-protein interaction analysis

All proteins labeled in *Figure 3e* plus Kif5a were input into STRINGdb (http://string-db.org) (*Szklarczyk et al., 2017*). Interactions sources included: textmining, experiments, databases, co-expression, neighborhood, gene fusion, and co-occurrence, with a medium 0.400 required interaction score. These data were exported into Cytoscape 3.7 for graphical organization.

### Statistical analysis

Data were analyzed using R version 3.5.0 (The R Foundation for Statistical Computing) or Graphpad Prism 7 (GraphPad, San Diego, CA). For correlation between proteomic samples, significance testing used Pearson's correlation coefficient in package 'GGally'. Quantitative proteomic comparisons of individual proteins were made using the 't-test' function of the 'limma' package in R. For statistical comparison of Kif5a transport after optic nerve crush by western blot, a one-sample, two-sided t-test was used. For all other comparisons between two samples, two-sided unpaired t-tests were used. For group comparisons, ordinary one-way ANOVA with post-hoc Sidak correction was used.

## Acknowledgements

This work was supported by the NIH: EY011261, EY027437, P30 EY019005, R01MH103134, and the Hahn Family Foundation to HTC, P41 GM103533 and R01MH067880 to JRY, P30 EY026877, the Glaucoma Research Foundation and Research to Prevent Blindness to JLG, and U01EY027261 to JLG, JRY, and HTC.

## Additional information

#### Funding

| Funder | Grant reference number | Author |
| --- | --- | --- |
| National Institutes of Health | EY011261 | Hollis T Cline |
| National Institutes of Health | EY027437 | Hollis T Cline |
| National Institutes of Health | P30 EY019005 | Hollis T Cline |
| National Institutes of Health | R01MH103134 | Hollis T Cline |
| National Institutes of Health | P41 GM103533 | John R Yates |
| Hahn Family Foundation | | Hollis T Cline |

| Funder | Grant reference number | Author |
|---|---|---|
| National Institutes of Health | R01MH067880 | John R Yates |
| National Institutes of Health | P30 EY026877 | Jeffrey L Goldberg |
| Glaucoma Research Foundation | | Jeffrey L Goldberg |
| Research to Prevent Blindness | | Jeffrey L Goldberg |
| National Institutes of Health | U01EY027261 | John R Yates Hollis T Cline Jeffrey L Goldberg |

The funders had no role in study design, data collection and interpretation, or the decision to submit the work for publication.

## Author contributions

Sahil H Shah, Conceptualization, Data curation, Formal analysis, Funding acquisition, Investigation, Methodology, Software, Validation, Visualization, Writing – original draft, Writing – review and editing; Lucio M Schiapparelli, Conceptualization, Data curation, Formal analysis, Investigation, Methodology, Validation, Writing – review and editing; Yuanhui Ma, Conceptualization, Data curation, Formal analysis, Investigation, Methodology, Resources, Visualization; Satoshi Yokota, Data curation, Formal analysis, Investigation, Methodology, Visualization; Melissa Atkins, Formal analysis, Investigation, Methodology, Visualization; Xin Xia, Investigation, Methodology, Validation; Evan G Cameron, Conceptualization, Investigation, Methodology; Thanh Huang, Data curation, Formal analysis, Investigation, Methodology, Resources, Visualization; Sarah Saturday, Formal analysis, Investigation, Methodology; Catalina B Sun, Data curation, Investigation, Methodology, Visualization; Cara Knasel, Investigation, Methodology, Visualization; Seth Blackshaw, Data curation, Project administration, Resources, Supervision, Visualization; John R Yates, Funding acquisition, Methodology, Project administration, Resources, Supervision, Validation; Hollis T Cline, Conceptualization, Funding acquisition, Methodology, Project administration, Resources, Supervision, Writing – review and editing; Jeffrey L Goldberg, Conceptualization, Formal analysis, Funding acquisition, Methodology, Project administration, Resources, Supervision, Writing – review and editing

## Author ORCIDs

Sahil H Shah (ID) http://orcid.org/0000-0002-6601-219X
Satoshi Yokota (ID) http://orcid.org/0000-0002-3727-7279
Seth Blackshaw (ID) http://orcid.org/0000-0002-1338-8476
John R Yates (ID) http://orcid.org/0000-0001-5267-1672
Hollis T Cline (ID) http://orcid.org/0000-0002-4887-9603
Jeffrey L Goldberg (ID) http://orcid.org/0000-0002-1390-7360

## Ethics

All animal experiments conformed to the ARVO statement for the Use of Animals in Ophthalmic and Vision Research and were reviewed and approved by the Institutional Animal Care and Use Committee (IACUC) and the Institutional Biosafety Committee of University of California, San Diego, Scripps Research, and Stanford University.

## Decision letter and Author response

Decision letter https://doi.org/10.7554/eLife.68148.sa1
Author response https://doi.org/10.7554/eLife.68148.sa2

---

# Additional files

## Supplementary files

• Transparent reporting form

## Data availability

All data generated during this study are included in the manuscript and supporting source files in excel format. Source data files have been provided for Figures 1, 4, 5 and Figure Supplement 2.

The following previously published datasets were used:

| Author(s) | Year | Dataset title | Dataset URL | Database and Identifier |
|---|---|---|---|---|
| Blackshaw S, Clark BS, Handa JT, Bremner R, Zack DJ | 2020 | scRNA-seq of the developing human retina | https://www.ncbi.nlm.nih.gov/geo/query/acc.cgi?acc=GSE138002 | NCBI Gene Expression Omnibus, GSE138002 |
| Hoang T, Wang J, Boyd P, Wang F, Hyde DR, Qian J, Blackshaw S | 2020 | Comparative transcriptomic and epigenomic analysis identifies key regulators of injury response and neurogenic competence in retinal glia | https://www.ncbi.nlm.nih.gov/geo/query/acc.cgi?acc=GSE135406 | NCBI Gene Expression Omnibus, GSE135406 |

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
