## [Editor Report]

This study uses unbiased proteomics to determine which proteins are transported along the axon transportome in the context of optic nerve injury. Kif5a is identified as the most significantly downregulated protein in the transportome after injury. Knocking out Kif5a resulted in degeneration of the axon, suggesting that Kif5a is crucial for maintaining a healthy optic nerve. Further, optic nerve analyses of the Kif5a KO vs the control upon injury identified defective mitochondrial transport and defective hRNP transport that has been identified as a cause of neurodegeneration in ALS upper motor neurons.

---

## [Decision Letter]

**Decision letter after peer review:**

Thank you for submitting your article "Quantitative transportomics identifies Kif5a as a major regulator of neurodegeneration" for consideration by *eLife*. Your article has been reviewed by 3 peer reviewers, and the evaluation has been overseen by a Reviewing Editor and Gary Westbrook as the Senior Editor. The following individuals involved in review of your submission have agreed to reveal their identity: Zhigang He (Reviewer #1); Judith A. Steen (Reviewer #3).

The reviewers have discussed their reviews with one another, and the Reviewing Editor has drafted what we consider the essential revisions to address to help you prepare a revised submission.

Essential revisions:

This study uses a proteomics-based method to quantify proteins that are transported along the axons and compare what happens to the "transportome" after optic nerve. The study identifies a number of downregulated proteins, among which Kif5a seems to be the most affected. The study thereafter used a number of approaches to investigate the role of Kif5a axon degeneration. The study is interesting and the methodology used in the study should be of value for the community, despite its limitation. Differential NHS-biotin uptake in the cell may in fact lead to bias in the protein labeling, with under representation of some proteins (e.g. Sncb and Arf3, described in the literature but not found in this study). Because of this limitation the authors should reanalyze their dataset and determine whether there are proteins that have been lost in their comparison, for example. Study limitations should also be discussed. Data should also explain better: for example, those related to Figure 5, which can lead to confusion.

1. The results in Figure 2e suggested "an equal amount of Kif5a in the retina with and without (crush) injury". However, Kif5a synthesis is reduced in a glaucoma model (Figure 2f). The authors should discuss if expression and transport of Kif5a are differentially affected in these injury models.

2. The proteomic data suggest specific regulation of Kif5a but not that of other Kifs by axotomy. Are axonal Kif subunit ratios altered after axotomy? A western blot of proximal optic nerve tissue should address this point.

3. The data related to the glaucoma model do not fit well with other experiments perhaps because the labeling method is different. Does AHA show axonal Kif5a change after axotomy? These data could be moved to Supplementary information unless the authors can expand their analysis.

4.Figure 2C and Figure 3 address distribution of protein and RNA, and seem somewhat redundant. It would be most useful to show mouse Kif5 immunohistology after ONC and in glaucoma. Human retinal immunohistology would add to the study more than the RNA profiles. The human data in Figure 3 are from one subject and would be better in a Supplemental Figure.

5. How many neurons and how many separate cultures were studied to obtain the kymograph data of Figure 5c-h. The "n" should represent separate cultures (not separate organelles in the same dish or same neuron).

6. In Figure 6, the optic nerve atrophies and loss of electrophysiological signals are expected but it is less clear why the ~50% reductions of Kif5a is important for cell survival. The authors should assess whether chronically deleted Kif5a heterozygous mice show progressive degeneration relative to WT.

7. It is striking that Kif5a is the only identified axon transport protein, in light of the involvement of multiple proteins in regulating this important process. Is it possible that other proteins are less abundant (below detection levels)? The authors should examine their dataset to assess this possibility and add the results to the discussion.

8. In Figure 2b, it is interesting to see some proteins' transport might be enhanced by injury. This seems a quite surprising finding as axonal injury has been usually linked to the down-regulation of axon transport. The authors should include some verification results and discuss possible mechanisms.

9. For the Kif5a knockout proteome profile in Figure 4c, is down regulation of Kif5a identified? If not, that raises questions about the method robustness. In addition, it would be best for the text and any network analysis (Figure 4d) to focus exclusively on significantly altered expression rather than non-significant changes. Both 14-3-3 proteins (Ywhab) and CRMP proteins (Dpysl5) have been implicated in axon growth previously.

10. The analysis in Figure 5a seems to show no correlation of KO and ONC responses. If the dots from all quadrants are considered, it would seem that there is no pattern at all, and a linear regression would be non-significant. This argues against a dominant role of Kif5a in crush response. Please discuss.*Reviewer #1 (Recommendations for the authors):*

In this manuscript, Shah et al., report their results of identifying Kif5a as a most affected protein of injury-induced axon transport in retinal ganglion cells (RGCs). The authors first describes a proteomics-based method (quantitative transportomics) of quantifying proteins that are synthesized in the cell bodies and transported along axon at different time points. With this method, the authors analyzed the effects of optic nerve injury on such axon transport and identified a list of most affected proteins. Among the injury-induced less transported proteins is Kif5a, which has been implicated in regulating anterograde axon transport. Further, the authors obtained additional evidence from a series of follow-up experiments to support the role of Kif5a as a critical regulator of neurodegeneration. Overall these results are largely convincing. The method described here and the obtained dataset should be valuable to the community.

*Reviewer #2 (Recommendations for the authors):*

This manuscript uses NHS-biotin injection into the vitreus followed by optic nerve profiling of biotin-labeled transported proteins with and without optic nerve crush, and reports that Kif5a is selectively down-regulated. This "transportome" profiling is a key strength of the study. A limitation of this approach is that the labeling method may tag proteins in a biased way that is based on NHS-biotin uptake, on protein subcellular localization and on Lys or N-terminus accessibility. Therefore, a number of proteins might be under-represented. In fact, the authors show reduced transport of Sncb and Arf3 as candidate proteins from the literature, but those proteins appear not to be detected in the profiling study.

The authors also profile the transportome of retinal ganglion cells after Kif5a conditional knockout without injury. Overlapping Kif5a KO protein changes with axotomy regulation suggest mitochondrial dysregulation. This observation leads to a characterization of mitochondrial axon transport in cultured RGC, showing reduced transport with Kif5a knockdown and greater with overexpression. This supports a role for Kif5a in organelle axon transport, though it is not clear whether this role is specific to certain cargoes versus others, nor whether there is a change in this Kif5a function after axotomy.

Finally, the authors delete Kif5a selectively in the retina and assess late time points. They observe a slow degeneration of retinal ganglion cells over several months. As RGCs are lost the optic nerve is atrophied and electrophysiological signals are lost. Thus, Kif5a is required for adult RGC survival, independent of damage. However, it is not clear whether axotomy-induced partial reductions of Kif5a are required for cell death induced by axonal trauma. In fact, preliminary data show that Kif5a overexpression is associated with slightly greater RGC loss after axotomy. Thus, the role of Kif5a protein changes in RGC neurodegeneration after axotomy remains unclear.*Reviewer #3 (Recommendations for the authors):*

The authors have performed logical experiments, and their data support their conclusions.

---

## [Author Response]

Essential revisions:1. The results in Figure 2e suggested "an equal amount of Kif5a in the retina with and without (crush) injury". However, Kif5a synthesis is reduced in a glaucoma model (Figure 2f). The authors should discuss if expression and transport of Kif5a are differentially affected in these injury models.

We thank the viewer for raising an excellent point. The reviewer is correct that in the immediate post-injury period, we found no difference in the abundance of Kif5a in the retina, but a difference in the transport of Kif5a to the axonal compartment. We then measured new protein synthesis in a chronic optic nerve injury model, which demonstrated a decrease in new Kif5a synthesis 3 weeks after the onset of injury. These orthogonal experiments demonstrate downregulation of Kif5a as part of an overall maladaptive response of RGCs to different types of injury at different time points, further highlighting the need to dissect the role of Ki5a in neurodegeneration. We now elaborate further on this in the Discussion.

2. The proteomic data suggest specific regulation of Kif5a but not that of other Kifs by axotomy. Are axonal Kif subunit ratios altered after axotomy? A western blot of proximal optic nerve tissue should address this point.

We thank the reviewer for noting one of our interesting findings, that through transport proteomics we are able to show how transport of Kif5a is specifically decreased out of the Kif family members we quantified. This particular question can be answered by the proteomic data presented, which we validated using several immunoprecipitation-western blots in figure 2a and 2d. While our Kif5b antibody was not successful in IP-western blots, we show a similar phenotype of Kif5b in the newly synthesized proteome in the glaucoma model in figure 2g, lending credence to the specificity seen in the optic nerve crush model. Additionally, as knockout of Kif5b did not have any affect on cellular survival as seen in Figure 6—figure supplement 1, we did not pursue this isoform further.

3. The data related to the glaucoma model do not fit well with other experiments perhaps because the labeling method is different. Does AHA show axonal Kif5a change after axotomy? These data could be moved to Supplementary information unless the authors can expand their analysis.

We chose to present this data as a second form of a more chronic axonal injury and used a protein labelling method that could isolate a similar time course of 24 hours as used in the transportomic studies. This new data was specific to cell body synthesis and did not measure transport. We are happy to either move this section to supplementary information as requested or keep in the main body as it pleases the editor.

4.Figure 2C and Figure 3 address distribution of protein and RNA, and seem somewhat redundant. It would be most useful to show mouse Kif5 immunohistology after ONC and in glaucoma. Human retinal immunohistology would add to the study more than the RNA profiles. The human data in Figure 3 are from one subject and would be better in a Supplemental Figure.

We appreciate the point raised by the reviewer, but respectfully disagree with this suggestion. Single-cell RNA sequencing of both human and mouse retina, even with a single sample of each, clearly shows the conserved expression patterns of Kif family members across the retina at a resolution not currently possible with immunostaining or western blotting. This experiment also demonstrates the specificity of *Kif5a* to RGCs in mice, a point not appreciable with immunohistology alone. While expression of Kif5a after injury is an interesting topic, the scope of our paper is focused on the transported fraction of Kif5a. Total Kif5a, which would be measured by immunohistology, is not the same as the transported fraction as demonstrated in figure 2d. On this basis, we have left this figure in the paper but can move to supplemental as the editor prefers.

5. How many neurons and how many separate cultures were studied to obtain the kymograph data of Figure 5c-h. The "n" should represent separate cultures (not separate organelles in the same dish or same neuron).

We now add this data to the figure legend of Figure 5. For reference, there were a total of 3 cultures of all conditions, with 18 neurons with the scramble shRNA plasmid, 13 neurons with the Kif5a overexpression plasmid, 22 neurons with the GFP plasmid, and 19 neurons with the Kif5a knockdown plasmid.

6. In Figure 6, the optic nerve atrophies and loss of electrophysiological signals are expected but it is less clear why the ~50% reductions of Kif5a is important for cell survival. The authors should assess whether chronically deleted Kif5a heterozygous mice show progressive degeneration relative to WT.

Indeed we do see degeneration in the heterozygous mice compared to wildtype, reminding the reviewer that this is not a heterozygote from birth with fewer RGCs. We thank the reviewer for this suggestion and agree that another interesting line of questioning is why heterozygous mice appear to demonstrate a partial phenotype similar to homozygous Kif5a knockout mice. This question has been explored previously in the Kif5a literature^1^, as neurodegeneration seen in hereditary spastic paraplegia typically involves heterozygous point mutations in one copy of *Kif5a* with a wildtype second copy. in vitro analysis of motor proteins with proportions of wildtype homodimers, mutant homodimers, and heterodimers found impaired global motor function in an autosomal dominant pattern. Although we use a deletion instead of a mutant motor, the imbalance of anterograde and retrograde motors may still result in the same phenotype. A separate, detailed study on cargo binding and transport in heterozygous knockout cells and animals would be an excellent idea. Nonetheless, it is clear from the current set of experiments that at the same time point analyzed, there is a dose-dependent effect of neurodegeneration. We also independently show progressive degeneration in homozygous Kif5a knockout mice. We have now added these changes to the Results and Discussions accordingly.

7. It is striking that Kif5a is the only identified axon transport protein, in light of the involvement of multiple proteins in regulating this important process. Is it possible that other proteins are less abundant (below detection levels)? The authors should examine their dataset to assess this possibility and add the results to the discussion.

This is an excellent point raised by the reviewer, and one that we now elaborate on in the Discussion. In proteomics, we are only able to quantify peptides above the limit of detection. By adding another layer of selection- biotinylation and pulldown of the relatively small portion of transported peptides in the optic nerve tissue compared to native optic nerve tissue- we add extra stringency to limit the false positive peptide detection. As mass spectrometry continues to improve, we will be able to sequence deeper into the transported proteome.

8. In Figure 2b, it is interesting to see some proteins' transport might be enhanced by injury. This seems a quite surprising finding as axonal injury has been usually linked to the down-regulation of axon transport. The authors should include some verification results and discuss possible mechanisms.

We want to emphasize that we find global axonal transport is decreased after injury, as seen in figure 1f, which is consistent with previous studies. When we quantified how the individual proteins that were transported compared to the global transport rate (which was decreased), we find examples of proteins whose relative rate compared to other transported proteins was not as affected, but are indeed expected to be still less than the uninjured condition. We have clarified this further in the Results section.

9. For the Kif5a knockout proteome profile in Figure 4c, is down regulation of Kif5a identified? If not, that raises questions about the method robustness. In addition, it would be best for the text and any network analysis (Figure 4d) to focus exclusively on significantly altered expression rather than non-significant changes. Both 14-3-3 proteins (Ywhab) and CRMP proteins (Dpysl5) have been implicated in axon growth previously.

Kif5a was not identified in all proteomic samples. Our conservative methodology does not allow for quantitative analysis if a peptide was not detected, so we chose to not include these proteins. For the ease of identification for the reader, we color-coded the significant proteins in figure 4c but still labelled the nonsignificant decreased proteins. We now color-coded the network analysis in Figure 4d to reflect the differences between these same proteins as listed in Figure 4c, however if the editor prefers removal of those proteins we are happy to change this figure.

10. The analysis in Figure 5a seems to show no correlation of KO and ONC responses. If the dots from all quadrants are considered, it would seem that there is no pattern at all, and a linear regression would be non-significant. This argues against a dominant role of Kif5a in crush response. Please discuss.

The reviewer is correct in the limited total correlation of both datasets. However, we asked a specific question in line with our initial data: “How does protein transport loss compare in these two data sets?” To address this question with our data, we only examined the lower left quadrant of this graph. We used an unbiased GO analysis on this quadrant to identify the significantly overrepresented cellular components seen in figure 5b. From this list, we applied previous literature of both axonal injury and Kif5a biology to hypothesize the role of mitochondrial transport as one such plausible mechanism connecting these injuries, but certainly not the only mechanism. As the role of axonal mitochondria have been extensively studied in optic nerve crush and glaucoma^2,3^ , we believe our in vitro manipulation of mitochondrial transport using Kif5a constructs demonstrates a plausible biological mechanism. Nevertheless, we now add this caveat to the Results.

References

1. Ebbing, B., Mann, K., Starosta, A., Jaud, J., Schöls, L., Schüle, R., and Woehlke, G. (2008). Effect of spastic paraplegia mutations in KIF5A kinesin on transport activity. Hum. Mol. Genet. *17*, 1245–1252.

2. Inman, D.M., and Harun-Or-Rashid, M. (2017). Metabolic Vulnerability in the Neurodegenerative Disease Glaucoma . Front. Neurosci. *11*, 146.

3. Coughlin, L., Morrison, R.S., Horner, P.J., and Inman, D.M. (2015). Mitochondrial morphology differences and mitophagy deficit in murine glaucomatous optic nerve. Investig. Ophthalmol. Vis. Sci. *56*, 1437–46.

4. A., L.S., S., A.F., C., K.S., Takbum, O., Anton, A., R., H.G., Irfan, S., Resy, C., Yingzi, Y., C-H., T.A., et al. (2011). Mutations in the RNA Granule Component TDRD7 Cause Cataract and Glaucoma. Science (80-. ). *331*, 1571–1576.

5. Schiapparelli, L.M., Mcclatchy, D.B., Liu, H., Sharma, P., Yates, J.R., and Cline, H.T. (2014). Direct Detection of Biotinylated Proteins by Mass Spectrometry. J. Proteome Res. *13*, 3966–3978.

6. Schiapparelli, L.M., Shah, S.H., Ma, Y., McClatchy, D.B., Sharma, P., Li, J., Yates, J.R., Goldberg, J.L., and Cline, H.T. (2019). The Retinal Ganglion Cell Transportome Identifies Proteins Transported to Axons and Presynaptic Compartments in the Visual System in vivo. Cell Rep.